# Deformable Butterfly: A Highly Structured and Sparse Linear Transform

**Rui Lin**[1,*]    **Jie Ran**[1,*]    **King Hung Chiu**[2]    **Grazinao Chesi**[1]    **Ngai Wong**[1,*]

[1] Department of Electrical and Electronic Engineering,
The University of Hong Kong, Hong Kong
Emial Address: {`linrui, jieran, chesi, nwong`}@eee.hku.hk
[2] United Microelectronics Centre (Hong Kong) Limited,
Hong Kong Science Park, N.T., Hong Kong
Emial Address: `khchiu@umechk.com`

## Abstract

We introduce a new kind of linear transform named **De**formable **But**terfly (DeBut) that generalizes the conventional butterfly matrices and can be adapted to various input-output dimensions. It inherits the fine-to-coarse-grained learnable hierarchy of traditional butterflies and when deployed to neural networks, the prominent structures and sparsity in a DeBut layer constitutes a new way for network compression. We apply DeBut as a drop-in replacement of standard fully connected and convolutional layers, and demonstrate its superiority in homogenizing a neural network and rendering it favorable properties such as light weight and low inference complexity, without compromising accuracy. The natural complexity-accuracy tradeoff arising from the myriad deformations of a DeBut layer also opens up new rooms for analytical and practical research. The codes and Appendix are publicly available at: https://github.com/ruilin0212/DeBut.

## 1 Introduction

The linear mapping in deep neural networks (DNNs) are mostly realized in the form of fully connected (FC) or convolutional (CONV) layers. These layers, together with feed-forward or feedback paths and nonlinear activations, then induce a plethora of neural architectures such as multilayer perceptrons (MLPs) [29, 23, 12], convolutional neural networks (CNNs) [11, 18, 31, 32], recurrent neural networks (RNNs) [3, 13, 30] and Transformers [33, 9], just to name a few. Although latest researches have designed and constructed highly capable networks (such as the GPT-3 [1]), the nature of DNNs still remains largely black-box and inaccessible due to its extreme nonlinearity. In fact, even the seemingly trivial linear transform can become rather non-trivial when structures are imposed. Prominent examples are the Fast Fourier Transform (FFT) and convolution operators which can be cast as linear mappings and formulated as matrix multiplications. Yet they are distinguished by their underlying butterfly and circulant structures, respectively, that act as strong structural priors and lead to very distinct behaviors.

Nonetheless, relatively little attention has been paid to deriving adaptable, or even better, learnable and structured fast linear transforms in DNNs. Among the works to compactly parametrize an FC layer in a DNN, a representative method is called the Fastfood Transform [20] that belongs to the category of kernel methods [4, 26] and employs random projection for feature learning. In particular, Fastfood uses Hadamard transform combined with diagonal Gaussian matrices to approximate a Gaussian random matrix. Adaptive Fastfood [34] further allows those diagonal matrices to become

---

*RL, JR, and NW contributed equally to this work.

learnable, thus achieving better approximation of an FC layer during backpropagation. Recently, learnable butterfly matrices are also proposed [5, 6] and demonstrated to be an effective FC substitute delivering high accuracy with much fewer parameters owing to the highly sparse butterfly factors. However, all these aforementioned schemes are restricted to powers-of-two (PoT) construction and thereby square structures, in which unequal input-output dimensions are handled simply by stacking square matrices followed by ad hoc dropping of inputs/outputs. Moreover, all these works aim at replacing only one or a few largest FCs in a neural network, which may not yield a significant compression in the overall number of weight parameters especially in modern networks having a majority of CNN layers. Even worse, the training time of such FC replacement may end up prohibitive (cf. Experimental Section) as constrained by the high-dimensional square matrices.

On the other hand, the proliferation of machine learning and artificial intelligence (AI) in recent years has spawned the era of edge AI whereby the DNN inference, and sometimes its training, are performed on edge devices with limited compute and storage resources. Numerous researchers have looked into compacting neural networks. In general, modern neural network compression techniques mainly fall into three categories, namely, low-rank matrix/tensor decomposition (e.g., [25, 16, 27]), weight and/or CNN filter pruning (e.g., [10, 22]) and low bitwidth quantization (e.g., [14, 28, 2, 24]). To this end, network compression via sparse and structured matrix factorization does not fall into any of these categories, and constitutes a new yet under-explored way of neural network compression. As revealed in [5], such structured and sparse matrix factor chains (such as the FFT butterflies) are often accompanied by fast inference due to their recursive nature and the availability of fast matrix-vector multiplication schemes [7].

Subsequently, this paper attempts to kill two birds (viz. learnable factorized linear transform with structured sparsity and flexible input-output sizes) with one stone by introducing a novel linear transform named deformable butterfly (DeBut) that generalizes the square PoT butterfly factor matrices [5, 6]. Specifically, a DeBut product chain (or simply a DeBut chain) can be sized to adapt to different input-output dimensions. For one thing, it does not limit itself to PoT blocks as in Fastfood Transform or butterfly matrices [20, 34, 5, 6]. Moreover, the intermediate matrix dimensions in a DeBut chain can either shrink or grow to permit a variable tradeoff between number of parameters and representation power. The flexibility of tuning the dense matrix sub-blocks in DeBut also permits an interpretation similar to the CNN receptive field for exploiting locality and correlation in data. In fact, as will be shown in experiments, a DeBut chain does not distinguish CONV or FC layers, and can be used as a substitute of both while maintaining a high output accuracy. To our knowledge, the DeBut linear transform is proposed for the first time, and this work is a starter to showcase its use in DNNs which we hope can provoke further theoretical and practical insights.

## 2 Background

### 2.1 Butterfly Matrix

DeBut is inspired by the learnable butterflies proposed in [5] and its subsequent Kaleidoscope matrix extension [6] (called a K-matrix) which is essentially a butterfly matrix multiplied to another transposed butterfly matrix. The learnability and efficacy of K-matrices are then demonstrated in several applications including: **1)** replacing handcrafted features in speech preprocessing and CNN channel shuffling; **2)** learning latent permutation to "unpermute" scrambled pixels in an input image; **3)** speeding up inference in the linear layers in a Transformer translation model. The schemes and examples in [5, 6], however, employ only the conventional square PoT butterflies and substitute only one or several big FC layers in a DNN.

To ease illustration, Fig. 1 shows the four butterfly factor matrices for a $16 \times 16$ case. Specifically, we adopt an information-flow viewpoint for explaining this butterfly hierarchy. To begin with, the rightmost input vector is partitioned into 16 individual entries. It can be seen that after going through the first matrix-vector product, every pair of entries are "mixed" by the $2 \times 2$ diagonal sub-blocks so that the output vector is partitioned in pairs meaning that, e.g., the top two entries now contain information from the top two individual entries in the original input vector and so on. Following this convention, one can see that the butterfly matrix product represents a progressive, stage-by-stage information mixer such that after the final matrix-vector product, each entry in the leftmost output carries the merged information from all individual entries of the rightmost input, thus corresponding to a fine-to-coarse-grained abstraction advancing from factor to factor through the butterfly chain.

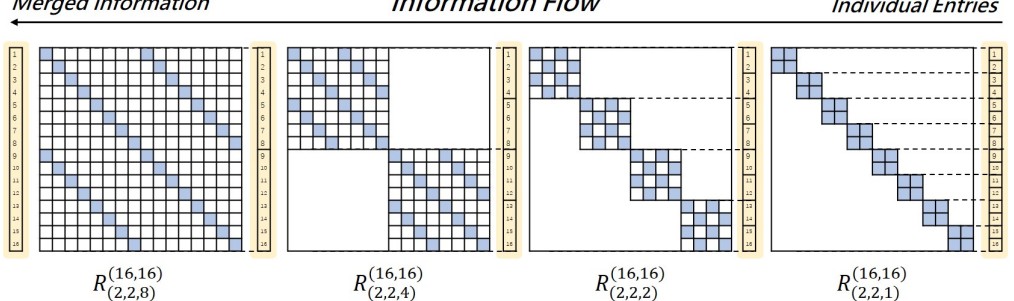

Figure 1: $16 \times 16$ Butterfly factor matrices and the hierarchical information flow from right to left, where the blue squares stand for nonzeros and the numbers in the vectors denote the positional indices. The dashed lines, which connect the DeBut factor and the vector, mark the entries that will be mixed up in the vector and the corresponding sub-block in the DeBut factor that works as the mixer. The partitions in the vectors denote the merging of information (cf. Section 2).

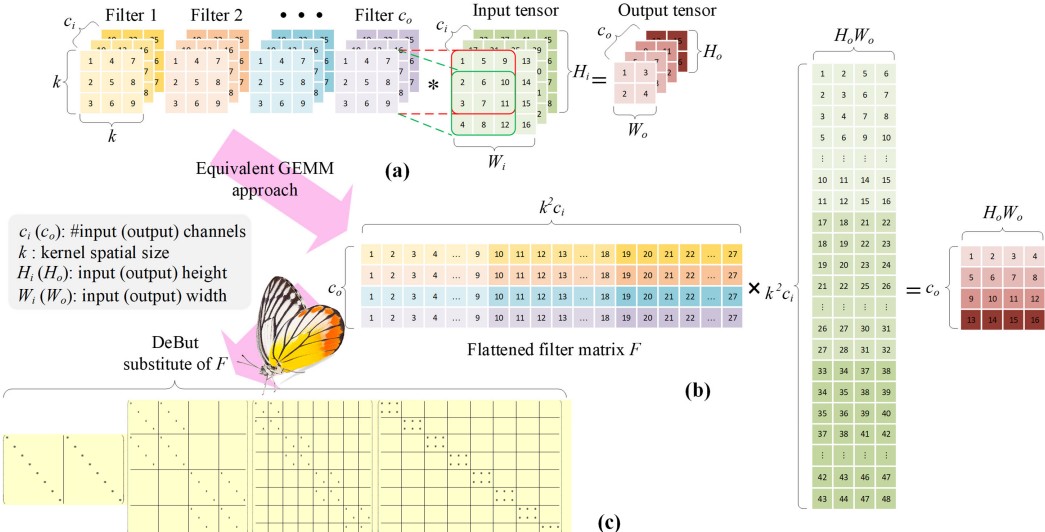

Figure 2: CNN convolution in its (a) conceptual, illustrative form; (b) equivalent matrix-matrix implementation by a flattened kernel matrix; (c) DeBut replacement of the kernel matrix.

## 2.2 Convolution as a Matrix Product

The convolution operation in a CNN is often conceptually depicted as a window or kernel sliding across the $c_i$-channel input feature, exemplified in Fig. 2(a). In practice, however, it is seldom realized this way due to the inefficient nested for-loop operations. Rather, it often proceeds by the highly optimized matrix-matrix product at the expense of larger memory footprint due to the extra data entries generated by the `im2col` command. Specifically, this command stretches the entries involved in each filter stride into a column and concatenates the columns into a flattened $k^2 c_i \times H_o W_o$ matrix, as depicted in Fig. 2(b). One interesting remark is that an FC layer is then equivalent to setting the kernel spatial size to be exactly the same as that of the input. Then, each output node in the FC corresponds to a one-step filtering (viz. without sliding/stepping) by the $H_i \times W_i \times c_i$ filter on the input tensor, as shown in Fig. 3. Consequently, both CONV and FC layers can be unified under this `im2col` notion whose difference lies only in the setting of the kernel spatial sizes yielding different formations of the corresponding flattened filter matrix $F$.

## 3 Deformable Butterflies

Building upon the success of butterfly-based FC layer approximation in [5, 6] (more precisely a PoT butterfly substitute since fine-tuning entails a newly learned layer instead of an approximation), the

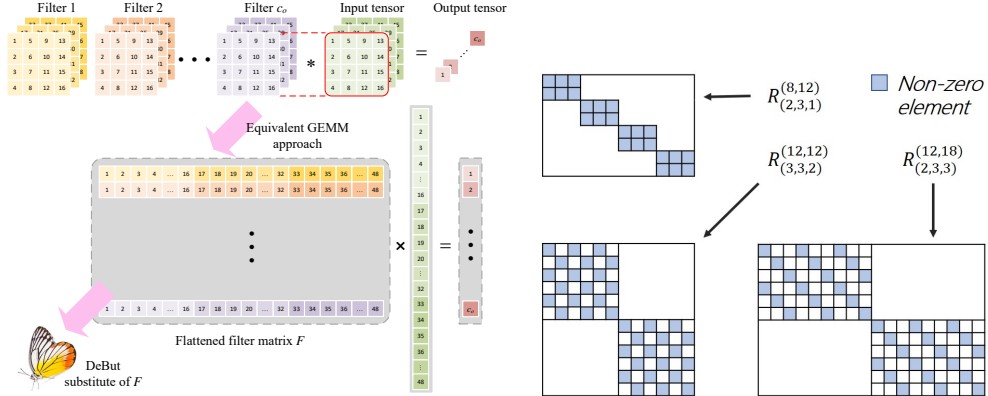

Figure 3: Convolution equivalent of FC.     Figure 4: Example DeBut factors.

central innovation and contribution of this work lie in the formulation of a DeBut chain that plays the role of $F$ in in Fig. 2(c) and subsequently that in Fig. 3. However, different from [5, 6], a DeBut chain is **1)** not necessarily PoT blocks or square; **2)** not intended as an approximate, but rather a replacement, to the circulant or doubly circulant structure adhering to CNN convolution; and **3)** not of a $\mathcal{B}\mathcal{B}^T$ construct ($\mathcal{B}$ being the set of butterfly matrices) as in the K-matrix [6] which doubles the amount of parameters to be learned.

We define the notion of a real-valued DeBut factor as $R_{(r,s,t)}^{(p,q)} \in \mathbb{R}^{p \times q}$ that contains block matrices along its main diagonal, wherein each block matrix is further partitioned into $r \times s$ blocks of $t \times t$ diagonal matrices. Fig. 4 shows several DeBut factor structures. This notational convention comes in handy to distill several important properties:

- Number of (not necessarily square) diagonal block matrices is $\frac{p}{rt}$ which also equals $\frac{q}{st}$;
- Number of parameters (blue squares in Fig. 4), also loosely called number of nonzeros though they can still be zeros by learning, is $ps$ which also equals $qr$.

With this notion in hand, the $16 \times 16$ butterfly factor matrix product in Fig. 1 can be expressed as

$$B = R_{(2,2,8)}^{(16,16)} R_{(2,2,4)}^{(16,16)} R_{(2,2,2)}^{(16,16)} R_{(2,2,1)}^{(16,16)}. \tag{1}$$

For brevity we also use the following shorthand for (1)

$$16 \xleftarrow[(2,2,8)]{} 16 \xleftarrow[(2,2,4)]{} 16 \xleftarrow[(2,2,2)]{} 16 \xleftarrow[(2,2,1)]{} 16. \tag{2}$$

A closer look reveals that in this standard butterfly chain the changing variable is the diagonal block size in PoT steps of 1, 2, 4 and 8. Another important observation of the butterfly factor chain is the progressive and hierarchical aggregation of information. This is reflected in the densification of intermediate products into dense diagonal blocks. Taking the standard butterflies in Fig. 1 and (1) for example, such densifying process, namely, producing $R_{(r,s,1)}^{(p,q)}$ with $r \times s$ dense sub-blocks, can be seen in the partial products in (1):

$$B = R_{(2,2,8)}^{(16,16)} \ R_{(2,2,4)}^{(16,16)} \ \underbrace{\underbrace{\underbrace{R_{(2,2,2)}^{(16,16)} R_{(2,2,1)}^{(16,16)}}_{R_{(4,4,1)}^{(16,16)}}}_{R_{(8,8,1)}^{(16,16)}}}_{R_{(16,16,1)}^{(16,16)}} \tag{3}$$

where the dense sub-blocks below the underbraces appear in the order of $R_{(2,2,1)}^{(16,16)}$, $R_{(4,4,1)}^{(16,16)}$, $R_{(8,8,1)}^{(16,16)}$ and $R_{(16,16,1)}^{(16,16)}$. In essence, such densification flow can be generalized to deformable blocks arising from the product of two contiguous DeBut factors, one with diagonal sub-blocks ($t > 1$) and another

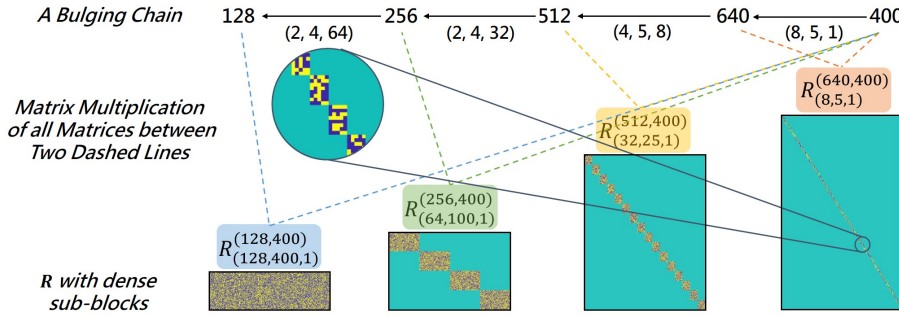

Figure 5: A bulging DeBut chain (not drawn to scale) and its densification process from right to left. The yellow, blue, and teal in the plots denote +1, -1 and 0, respectively.

with dense sub-blocks ($t = 1$), say $R^{(p_2,q_2)}_{(r_2,s_2,t_2)} R^{(p_1,q_1)}_{(r_1,s_1,1)}$ where $t_2 > 1$. It can be further shown that such densifying product mandates $q_2 = p_1$ and $t_2 = r_1$, leading to:

$$R^{(p_2,q_1)}_{(r_2r_1,s_2s_1,1)} \leftarrow R^{(p_2,p_1)}_{(r_2,s_2,r_1)} R^{(p_1,q_1)}_{(r_1,s_1,1)}. \tag{4}$$

Next, we analyze the complexity of evaluating a DeBut product chain versus the matrix-matrix multiply in Fig. 2(b). We emphasize that such a comparison is made from a purely mathematical viewpoint without considering specific hardware optimization. Specifically, when given a CONV layer with a 4-D kernel tensor of size $[c_i, c_o, k, k]$ and an input feature tensor of size $[c_i, H_i, W_i]$, the complexity of convolution via General Matrix Multiply (GEMM) is $\mathcal{O}(c_o \cdot k^2 c_i \cdot H_o W_o)$, while that of DeBut is $\mathcal{O}(\max_{i \in \{1, \cdots, N\}} p_i s_i \cdot H_o W_o)$, where $N$ is the number of DeBut factors, and $\max_{i \in \{1, \cdots, N\}} p_i s_i$ denotes the number of nonzeros in the DeBut factor containing the most nonzeros. It is worth noting that evaluating a DeBut chain has much lower complexity compared with GEMM, attributed to the exponentially fewer total number of nonzeros from all DeBut factors. Detailed expositions of the densification condition and additional complexity analysis can be found in Appendix I.

### 3.1 Designing DeBut Factors

Now we are ready to present the core step of replacing the matrix $F \in \mathbb{R}^{c_{out} \times k^2 c_{in}}$ by an appropriately sized DeBut chain. Again, the best way is to illustrate through a number example. For this purpose, we choose the biggest FC layer in a modified LeNet (cf. Table 1) wherein the flattened $F$ measures $128 \times 400$. Apparently, in going from the input to output, the input vector dimension (400) has to be shrunk to that of the output (128). In this regard, there multiple choices for the DeBut chains as long as **1)** the "boundary" dimensions are met; and **2)** the densifying condition in (4) can be satisfied throughout the chain (see (3)) so that the final product $B$ does not contain any "voids" (zeros), or else some input entries will be nullified causing incomplete information flow.

In particular, because of the deformable nature of a DeBut factor, we can devise monotonic (monotonically shrinking or expanding) or bulging (expand-then-shrink) chains that feature different number of parameters and accuracies. For example, Fig. 5 shows a bulging chain and its corresponding densification process. In this example, we set every nonzero to be random $+1$ or $-1$, i.e, bipolar. Then, it can be traced through the chain and verified that every dense block in the product accumulation (including the leftmost final product in Fig. 5) is also bipolar. We call this a "bipolar test" for checking the integrity of information flow, which essentially means that no entry in the linear transform is forced zero by an ill-posed DeBut chain. Analogous to the myriad of choices in CNN filters, we solicit insights and guidelines for picking effective DeBut chains in the experimental section.

### 3.2 Initializing the DeBut Factors

If we have a pretrained network whose $F$ matrix (see Fig. 2(b)) is already available, then it is possible to initialize the DeBut factors by approximating their product to $F$. Here we describe an alternating least squares (ALS) scheme to initialize these factors such that their product stays close to $F$ in the 2-norm sense. Specifically, for a given $F$ and starting with a randomly initialized DeBut chain

| LeNet | Weights | | Outputs | VGG | Weights | | Outputs |
|---|---|---|---|---|---|---|---|
| | $[c_i, c_o, k, k]$ | $[c_o, k^2 c_i]$ | $[c_o, H_o, W_o]$ | | $[c_i, c_o, k, k]$ | $[c_o, k^2 c_i]$ | $[c_o, H_o, W_o]$ |
| CONV1 | [1, 8, 3, 3] | [8, 9] | [8, 26, 26] | CONV1 | [3, 64, 3, 3] | [64, 27] | [64, 32, 32] |
| CONV2 | [8, 16, 3, 3] | [16, 72] | [16, 11, 11] | CONV2 | [64, 64, 3, 3] | [64, 576] | [64, 32, 32] |
| FC1 | [16, 128, 5, 5] | [128, 400] | [128, 1, 1] | CONV3 | [64, 128, 3, 3] | [128, 576] | [128, 16, 16] |
| FC2 | [128, 64, 1, 1] | [64, 128] | [64, 1, 1] | CONV4 | [128, 128, 3, 3] | [128, 1152] | [128, 16, 16] |
| FC3 | [64, 10, 1, 1] | [10, 64] | [10, 1, 1] | CONV5 | [128, 256, 3, 3] | [256, 1152] | [128, 16, 16] |
| | | | | CONV6 | [256, 256, 3, 3] | [256, 2304] | [256, 8, 8] |
| | | | | CONV7 | [256, 256, 3, 3] | [256, 2304] | [256, 8, 8] |
| | | | | CONV8 | [256, 512, 3, 3] | [512, 2304] | [256, 8, 8] |
| | | | | CONV9 | [512, 512, 3, 3] | [512, 4608] | [512, 4, 4] |
| | | | | CONV10 | [512, 512, 3, 3] | [512, 4608] | [512, 4, 4] |
| | | | | CONV11 | [512, 512, 3, 3] | [512, 4608] | [512, 4, 4] |
| | | | | CONV12 | [512, 512, 3, 3] | [512, 4608] | [512, 4, 4] |
| | | | | CONV13 | [512, 512, 3, 3] | [512, 4608] | [512, 4, 4] |
| | | | | FC1 | [512, 512, 1, 1] | [512, 512] | [512, 1, 1] |
| | | | | FC2 | [512, 10, 1, 1] | [512, 10] | [10, 1, 1] |

Table 1: (Left) The modified LeNet with a baseline accuracy of 99.29% on MNIST. (Right) VGG-16-BN with a baseline accuracy of 93.96% on CIFAR-10. In both networks, the activation, max pooling and batch normalization layers are not shown for brevity.

wherein one of the factors is $M$, we first multiply all factors on the left of $M$ and lump them into $L$, and similarly for $R$ on the right. If $M$ is on the left/rightmost, then we assume an identity matrix to its left/right. Now, we use a toy example for the ALS illustration, whose generalization is straightforward. Note that all notations in this subsection are local and not to be confused with other sections. Suppose we have a $4 \times 4$ $M$ with eight $m_{ij}$'s in the example below,

$$ F = L \left( \begin{array}{cc|cc} m_{11} & 0 & m_{13} & 0 \\ 0 & m_{22} & 0 & m_{24} \\ \hline m_{31} & 0 & m_{33} & 0 \\ 0 & m_{42} & 0 & m_{44} \end{array} \right) R. \tag{5} $$

Using Matlab-style notation for a row/column, simple algebra shows $F$ can be broken into eight rank-1 factors $L_{:i} m_{ij} R_{j:}$. Utilizing the Kronecker product property $\text{vec}(L_{:i} m_{ij} R_{j:}) = R_{j:}^T \otimes L_{:i} m_{ij}$, we then have the LS approximation of $m_{ij}$'s being the solution of the linear system

$$ \text{vec}(F) = \left( R_{1:}^T \otimes L_{:1} \quad \cdots \quad R_{4:}^T \otimes L_{:4} \right) \begin{pmatrix} m_{11} \\ m_{31} \\ \vdots \\ m_{44} \end{pmatrix}. \tag{6} $$

We note the matrix preceding the unknown vector is a Khatri–Rao product (i.e. column-wise Kronecker). Once $M$ is solved, then we advance to its adjacent factor and repeat the same procedure. In short, ALS entails sweeping the chain back and forth until the residual error no longer decreases. In our experiments, the ALS consistently converges within only a few sweeps.

## 4 Experiments

We test the proposed DeBut chains on LeNet (Table 1, left), VGG-16-BN (Table 1, right) and ResNet-50 [11] using the standard MNIST [21], CIFAR-10 [17] and ImageNet [8] datasets, respectively. The results are presented in the subsections below. Since DeBut is a superset of the (square) Butterfly and Kaleidoscope matrices in [5, 6], we do not repeat their specific applications. Rather, our primary goal is to validate the use of DeBut chains in substituting FC and CONV layers in a DNN, and to borne out important insights and design guidelines. In Section 4.4, we contrast DeBut with its closest and most practical linear transform work named Adaptive Fastfood [20]. The results further demonstrate the superiority of DeBut.

**Implementation details.** In our experiments, we compare the results of DeBut factors with and without ALS initializations. Note that ALS is only applied once prior to training, so as to initialize a DeBut chain as an approximate of a pretrained layer (viz. $F$), if the latter is available. We set the number of sweeps equal to 5 when initializing relatively small layers (viz. all layers in LeNet, and CONV1~3 and FC1 in VGG-16-BN). Whereas the number of sweeps is set to 10 for larger layers in VGG-16-BN and ResNet-50. The convergence speed of ALS initialization is presented in Appendix II (Fig. A1). When training the neural networks after substituting the selected layers by DeBut chains, we use the standard stochastic gradient descent (SGD) for fine-tuning. We remark that

the layers that are not substituted by DeBut factors have pretrained parameters instead of randomly initialized ones. On MNIST and CIFAR10 datasets, the learning rate is 0.01 with a decaying step of 50, the batch size and the number of epochs are set to 64 and 150, respectively. As for ImageNet training, the decaying step is 30 and the training warms up in the first 5 epochs. The batch size and the number of epochs are 128 and 100, respectively. All coding is done with PyTorch, and experiments run on an NVIDIA GeForce GTX1080 Ti Graphics Card with 11GB frame buffer. This also shows DeBut networks can be readily trained using very decent resources.

## 4.1  LeNet Trained on MNIST

We first test the proposed DeBut structure on the modified LeNet shown in Table 1 (left column) whose baseline accuracy is 99.29%. As described in Section 2.2, a FC layer can be regarded as a CONV layer with its kernel spatial size equal to that of the input. To quickly zoom into the benefits of DeBut, we pick the two biggest FC layers ([128,400] and [64,128]) and the largest CONV2 layer ([16,72]) and substitute them with DeBut chains. We devise monotonic chains with both fast and slow shrinkage, as well as a few bulging chains with different bulge sizes and shapes. Their results are listed in Table 2 which shows DeBut plug-in for three cases: FC1 only, CONV2 only and CONV2+FC1+FC2. We define layer-wise compression (LC) and model-wise compression (MC) as the amount of reduced (i.e. zeroed) parameters in a DeBut layer with respect to the number of parameters in the original layer and the number of parameters in the whole network, respectively. When computing LC, we take account of the local FC or CONV layer only, in which bias are included. As for the MC metric, the global model is considered, including parameters of bias and BN layers. Apparently, the higher the LC or MC the better. Additional experiments using different monotonic and bulging chains are provided in Appendix III (Tables A1 and A2).

We note that a CONV layer is already a significant parametric reduction from an FC layer, yet DeBut is still able to reduce the number of parameters further while delivering promising output accuracy. In this small example, ALS initialization is not always beneficial for learning the latent information, but its advantages will become obvious in larger examples to follow. Interestingly, this echoes the unimpressive results of Adaptive Fastfood vs the original (non-adaptive) Fastfood in small examples, in which the former excels significantly in larger models [34].

| Layer | Monotonic/Bulging Chains | LC | MC | Params | Acc% (no ALS) | Acc% (w/ ALS) |
|---|---|---|---|---|---|---|
| FC1 | $128 \xleftarrow[(1,2,32)]{} 256 \xleftarrow[(2,2,16)]{} 256 \xleftarrow[(16,25,1)]{} 400$ $128 \xleftarrow[(2,2,64)]{} 128 \xleftarrow[(2,2,32)]{} 128$ | 85.00% | 70.78% | $17.96K$ | $98.89(\pm0.08)$ | $98.72(\pm0.02)$ |
| CONV2 | $16 \xleftarrow[(2,2,8)]{} 16 \xleftarrow[(2,6,4)]{} 48 \xleftarrow[(1,2,4)]{} 96 \xleftarrow[(4,3,1)]{} 72$ | 41.67% | 1.04% | $60.84K$ | $99.02(\pm0.06)$ | $98.86(\pm0.02)$ |
| CONV2 | $16 \xleftarrow[(2,6,8)]{} 48 \xleftarrow[(1,2,8)]{} 96 \xleftarrow[(2,2,4)]{} 96 \xleftarrow[(4,3,1)]{} 72$ | 41.67% | | | | |
| FC1 | $128 \xleftarrow[(1,2,32)]{} 256 \xleftarrow[(2,2,16)]{} 256 \xleftarrow[(16,25,1)]{} 400$ $128 \xleftarrow[(2,2,64)]{} 128 \xleftarrow[(2,2,32)]{} 128$ | 85.00% | 83.43% | $10.19K$ | $98.64(\pm0.15)$ | $97.27(\pm0.02)$ |
| FC2 | $64 \xleftarrow[(2,2,8)]{} 64 \xleftarrow[(2,2,4)]{} 64 \xleftarrow[(2,2,2)]{} 64 \xleftarrow[(2,4,1)]{} 128$ $64 \xleftarrow[(2,2,32)]{} 64 \xleftarrow[(2,2,16)]{} 64$ | 89.06% | | | | |

Table 2: DeBut substitution of single and multiple layers in the modified LeNet. LC and MC stand for layer-wise compression and model-wise compression, respectively, whereas "Params" means the total number of parameters in the whole network. These notations apply to subsequent tables.

## 4.2  VGG Trained on CIFAR-10

After verifying the efficacy of DeBut layers in the small LeNet example, we move on to a larger network. Specifically, we train a VGG-16-BN network (Table 1, right) on CIFAR-10 which is a variant of the original VGGNet [22] and has a baseline accuracy of 93.96%. We first substitute the last CONV layer (CONV13) with a DeBut layer to quickly see its effect. As shown in Appendix IV (Table A3), three monotonic and three bulging DeBut chains are designed, initialized either randomly or with ALS, to test their performance. All the chains achieve remarkable LCs in CONV13, with only little drop in accuracy when using ALS initialization. Besides, one of the monotonic chains, listed in Table 3, has an MC of 15.23% while achieving a prediction accuracy of 93.91% which is very close to the baseline.

Recalling from Table 1, CONV9~13 have an $F$ of the same size $[512, 4608]$. Apparently, it is time-consuming to trial different DeBut chains in each layer and pick the best chain combination across multiple CONV layers. Instead, we select a proper Debut chain structure and assign it to layers of the same size. Amid our tests of different chains in the same layer (Appendix IV, Tables A4 & A5), we have observed that a bulging chain will generally yield a smaller ALS error, attributed to its bigger number of parameters. However, bulging chains with more nonzeros may not necessarily lead to better (re)trained accuracy than their monotonic counterparts. To the contrary, the latter tend to behave more stably in the training process and can often achieve a higher final accuracy. Subsequently, we deploy a monotonic chain with the least ALS error to obtain a high compression and accuracy simultaneously. In Table 3, all CONV layers with 512 output channels (CONV8~13) are replaced with DeBut monotonic chains. Amazingly, this VGG-16-BN with DeBut layers achieves a remarkable MC of $83.77\%$ with only a slight $0.24\%$ accuracy drop.

| Layer | Chain(s) | LC | MC | Params | Acc% (w/ ALS) |
|---|---|---|---|---|---|
| CONV13 | $4096 \xleftarrow{(2,2,16)} 4096 \xleftarrow{(2,2,8)} 4096 \xleftarrow{(8,9,1)} 4608$ 
 $512 \xleftarrow{(2,4,256)} 1024 \xleftarrow{(2,4,128)} 2048 \xleftarrow{(2,4,64)} 4096 \xleftarrow{(2,2,32)}$ | 96.79% | 15.23% | $12.71M$ | $93.91(\pm0.08)$ |
| CONV8 | $2048 \xleftarrow{(2,2,32)} 2048 \xleftarrow{(2,2,16)} 2048 \xleftarrow{(2,2,8)} 2048 \xleftarrow{(8,9,1)} 2304$ 
 $512 \xleftarrow{(2,2,256)} 512 \xleftarrow{(2,4,128)} 1024 \xleftarrow{(2,4,64)}$ | 96.79% | 83.77% | $2.43M$ | $93.72(\pm0.07)$ |
| CONV9~13 | $4096 \xleftarrow{(2,2,16)} 4096 \xleftarrow{(2,2,8)} 4096 \xleftarrow{(8,9,1)} 4608$ 
 $512 \xleftarrow{(2,4,256)} 1024 \xleftarrow{(2,4,128)} 2048 \xleftarrow{(2,4,64)} 4096 \xleftarrow{(2,2,32)}$ | 96.79% | | | |

Table 3: DeBut substitution of single and multiple layers in VGG-16-BN.

## 4.3 ResNet-50 Trained on ImageNet

Next, we examine the effectiveness of DeBut layers using a ResNet-50 trained on ImageNet which is a much larger and more complicated dataset than both MNIST and CIFAR-10. In particular, we use DeBut factors to substitute the CONV layers (9 in total) in the last three bottleneck blocks. The chain details in each layer are described in Appendix V (Tables A6 & A7). As an ablation study, we employ two sets of chains, one containing only bulging chains (DeBut-bulging) whereas another contains only monotonic chains (DeBut-mono). The two sets of chains have different properties. For the set of bulging chains, their ALS errors are smaller and therefore can approximate the layer-wise $F$ more accurately. However, bulging chains have lower compression due to additional parameters compared with monotonic chains.

The results are listed in Table 4. For DeBut-bulging, the number of parameters reduces by $47.56\%$, the top-1 accuracy is $74.52\%$, $1.49\%$ lower compared with the baseline of $76.01\%$. For DeBut-mono, the compressed model has $0.3M$ fewer parameters than the DeBut-bulging model, yet still achieving a comparable $74.34\%$ top-1 accuracy.

| Model | MC | Params | Top-1(%) with ALS | Top-5(%) with ALS |
|---|---|---|---|---|
| ResNet-50 | —— | $25.55M$ | 76.01 | 92.93 |
| DeBut-bulging | 47.56% | $13.40M$ | 74.52 | 92.18 |
| DeBut-mono | 48.74% | $13.10M$ | 74.34 | 92.31 |

Table 4: Results of ResNet-50 on ImageNet. DeBut chains are used to substitute the CONV layers in the last three bottleneck blocks. The DeBut chains used are described in Appendix V (Tables A6 & A7).

## 4.4 Comparison

### 4.4.1 DeBut vs. Other Linear Transform Schemes

We further contrast DeBut against the original Butterfly [5] and Adaptive Fastfood [34] on MNIST and CIFAR10. The comparison on the ImageNet dataset is not included since Fastfood training is impractically time-consuming on large datasets. The results reported of Butterfly are our implementation using their officially released codes. As for Adaptive FastFood, we obtained the results by

modifying the official codes of Fastfood available in sklearn (we remark that the sklearn codes are the fastest implementation we could find). Therefore, the comparison is fair as all algorithms are compared in the standard coding and software environment.

In Table 5, we do both single-layer and three-layer replacement on the modified LeNet and apply different economic linear transforms. Although Adaptive Fastfood has both good accuracy and high compression, it suffers from two major limitations. First, this method only supports PoT input-output sizes. But even more restrictive is the computational complexity of Adaptive Fastfood that scales at $\mathcal{O}(n \log n)$ where $n$ is the dimension of the input vector. As discussed in Appendix I, the complexity of DeBut is $\mathcal{O}(N \cdot \max_{i=\{1, \cdots, N\}} q_i r_i)$ where $N$ is the number of factors in a DeBut chain. When replacing FC and CONV layers, $r_i$ is a small number, $N \leq \log n$ and $q_i \leq n$ since we do not expand the input size to a larger number than $2^{\lceil \log n \rceil}$. Subsequently, the complexity of DeBut is lower than $\mathcal{O}(n \log n)$. Additionally, large batches of data are needed in training, for which Adaptive Fastfood performs repetitive Fast Hadamard Transform (FHT) while DeBut only needs tensor element-wise multiplication. In practice, the FHT is exceedingly time-consuming and makes it difficult to train Adaptive Fastfood on multiple layers in a large CNN.

Table 6 showcases the effectiveness of DeBut versus the other two methods. It can be seen that DeBut achieves the highest prediction accuracy with only $0.3M$ more parameters. We also note that Adaptive Fastfood takes around 2100s for each training epoch, making its training prohibitively slow even just for CIFAR-10.

| Layer | Method | MC | Params | Acc% |
|---|---|---|---|---|
| FC1 | Adaptive Fastfood | 80.78% | $11.82K$ | $98.89(\pm 0.07)$ |
| | Butterfly | 68.29% | $19.50K$ | $98.64(\pm 0.09)$ |
| | DeBut | 70.78% | $17.96K$ | $98.89(\pm 0.08)$ |
| CONV2 & FC1 & FC2 | Adaptive Fastfood | 94.73% | $3.2K$ | $98.61(\pm 0.08)$ |
| | Butterfly | 79.75% | $12.45K$ | $98.02(\pm 0.17)$ |
| | DeBut | 83.43% | $10.19K$ | $98.64(\pm 0.15)$ |

Table 5: Comparison results for LeNet on MNIST. For DeBut, the chains for the corresponding layers are the same as in Table 2.

| Layer | Method | MC | Params | Acc% | Training Time(s/epoch) | Inference Time(s) |
|---|---|---|---|---|---|---|
| CONV8~13 | Adaptive Fastfood | 85.65% | $2.15M$ | $93.60(\pm 0.02)$ | 2100 | 148.27 |
| | Butterfly | 85.82% | $2.13M$ | $93.34(\pm 0.12)$ | 105 | 4.58 |
| | DeBut | 83.77% | $2.43M$ | $93.72(\pm 0.07)$ | 50 | 4.01 |

Table 6: Comparison results for VGG-16-BN on CIFAR10. For DeBut, the chains for the corresponding layers are the same as in Table 3.

### 4.4.2 DeBut vs. Conventional Compression Schemes

Besides comparing DeBut with its closest schemes (i.e., Butterfly [5] and Adaptive Fastfood [34]) in Section 4.4.1, we also discuss the relations between DeBut and the popular compression schemes, namely, quantization, pruning and low-rank decomposition to further benchmark the differences between DeBut and existing approaches.

| Layer | Method | Params | Acc(%) |
|---|---|---|---|
| CONV8~13 | Baseline | $14.99M$ | 93.96 |
| | Tucker-2 | $3.21M$ | 93.36 |
| | DeBut | $2.43M$ | 93.71 |

Table 7: Comparison results between DeBut and Tucker-2 for VGG-16-BN on CIFAR10. The chains for DeBut layers are the same as in Table 3.

For weight quantization and pruning, we stress that DeBut is orthogonal and complementary to them, which means DeBut can be readily plugged in to compress further the compact models obtained by them. Therefore, we do not compare DeBut with quantization and pruning.

When compared with low-rank decomposition, it is worth noting that all matrix factors in a given DeBut chain are full-rank, and therefore the chain product as well. In addition, none of the tensor decomposition approaches (e.g., tensor train, Tucker, CP decomposition, etc.) show low-rank structures in DeBut. In short, DeBut is a brand-new way of structural matrix factorization with high sparsity and does not belong to the class of low-rank matrix factorization. In Table 7, we compare

DeBut with a classic low-rank decomposition method called Tucker-2 [15], it can be observed that DeBut achieves better performance than Tucker-2 (93.71% vs. 93.36%) with even fewer number of parameters (2.43M vs. 3.21M). This result demonstrates that DeBut has comparable or even better compression ability than low-rank decomposition approaches.

## 5   Additional Remarks

A few important remarks are in order:

- DeBut layers are truly practical and can be realized for ImageNet-scale datasets and networks using standard resources, where other fast linear transform schemes fail due to prohibitive training and inference times.

- Both Winograd [19] and FFT are algorithmic-level acceleration of CONV operation, and are mathematically equivalent to convolution. The methods in [5, 6] are also fitting butterfly structures onto the circulant matrices or FFT butterflies corresponding to CNN convolution. To this end, DeBut is *fundamentally different* as it is neither a reformulation nor an approximate of CONV, but a brand new structural regularizer of its own.

- A beauty of DeBut layers is that they further unify and homogenize FC and CONV layers in the sense that DeBut is doing this linear mapping in its unique butterfly-like attribute that alleviates the PoT limitation with reduced complexity than dense matrix multiplication. Indeed, CONV and FC only differ in their ways of aggregating information which is analogous to sizing the receptive field in a CNN kernel, as shown in Figs. 2 & 3. This maneuver can also be controlled by juggling the dense block size in the rightmost DeBut factor in a chain.

- The reason for having both types (bulging and monotonic) of DeBut chains is threefold: compression ability, performance, and stability. Although a monotonic chain can obtain a more compact and robust model, a bulging chain is expected to have better performance since in principle it has a higher representation power and may learn better latent information. Therefore, we need both types of chains to meet different model requirements. Since this article's key innovation is the new DeBut structured matrix factorization and not the enumeration of chains, we will provide more chain designing details in our upcoming work.

- The progressive, hierarchical flow in the DeBut factor product chain provides an important implication for a pipelined DNN inference speedup, namely, the linear mapping is broken into multiple phases of granular (and cheap) matrix products whose intermediate results can be pipelined for an overall speedup. This will be pursued in our future work.

## 6   Conclusion

This work literally thinks out of the (square) box to introduce a new class of unnecessarily square, deformable butterfly (DeBut) factors that are verified to be economic substitute of CONV and FC layers without sacrificing accuracy. The inherent structured sparsity in a DeBut chain naturally gives rise to a fine-to-coarse-grained hierarchical mapping, as well as a versatile way to achieve network compression. Experiments have shown the superiority of DeBut linear transform over competing algorithms, especially for large networks where the training and inference complexities matter. It is expected more interesting theories and practical insights will arise when the DeBut layer is coupled with other application-specific networks and/or optimization schemes.

## Acknowledgement

This work is supported in part by the General Research Fund (GRF) project 17206020, and in part by the in-kind support of United Microelectronics Centre (Hong Kong) Limited.

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
