# Supplementary Materials of DeBut

**Rui Lin**[1,*]    **Jie Ran**[1,*]    **King Hung Chiu**[2]    **Grazinao Chesi**[1]    **Ngai Wong**[1,*]

[1] Department of Electrical and Electronic Engineering,
The University of Hong Kong, Hong Kong
Emial Address: {`linrui, jieran, chesi, nwong`}@eee.hku.hk
[2] United Microelectronics Centre (Hong Kong) Limited,
Hong Kong Science Park, N.T., Hong Kong
Emial Address: {`kuchiu`}@umechk.com


## Appendix I: Complexity of DeBut Multiplication

In this section, we analysis the complexity of DeBut in detail when given a CONV layer with weights $[c_i, c_o, k, k]$ and an input $[c_i, H_i, W_i]$. We use $K \in \mathbb{R}^{c_o \times c_i \cdot k \cdot k}$ to denote the flattened weights, and assume it can be represented by a series of DeBut factors $R_{(r_i, s_i, t_i)}^{(p_i, q_i)} = (i = 1, \cdots, N)$.

The number of nonzero elements in $R_{(r_i, s_i, t_i)}^{(p_i, q_i)}$ is $\frac{p_i}{r_i \cdot t_i} \times r_i \cdot s_i \cdot t_i = p_i \cdot s_i$, or equivalently as $\frac{q_i}{s_i \cdot t_i} \times r_i \cdot s_i \cdot t_i = q_i \cdot r_i$. By using $X$ to denote the corresponding input matrix for $K$, the convolution process can be denoted as

$$K \cdot X = E_1 \cdot \prod_{i=0}^{N-1} R_{(r_{(N-i)}, s_{(N-i)}, t_{(N-i)})}^{(p_{(N-i)}, q_{(N-1)})} \cdot E_2 \cdot X, \tag{A1}$$

where $E_1$ and $E_2$ are identity matrices of size $[c_o, c_o]$ and $[k^2 c_i, k^2 c_i]$, respectively.

For easier understanding, the input $X$ can be regarded as $H_o W_o$ columns of length $k^2 c_i$ (cf. Fig. 2). Since $E_1$, $E_2$ and every $R_{(r_i, s_i, t_i)}^{(p_i, q_i)}$ $(i = 1, \cdots, N)$ are sparse, the naive sparse matrix-vector multiplication algorithm can be employed. As he number of nonzero elements in $E_1$ and $E_2$ are $c_o$ and $k^2 c_i$, respectively, the corresponding required matrix-vector operation for them are $\mathcal{O}(c_o)$ and $\mathcal{O}(k^2 c_i)$, which reflect that the complexity cost will be dominated by the DeBut factors.

For each $R_{(r_i, s_i, t_i)}^{(p_i, q_i)}$, the number of nonzeros is $p_i s_i$ $(q_i r_i)$. Therefore, the matrix-vector multiplication required by each $R_{(r_i, s_i, t_i)}^{(p_i, q_i)}$ is $\mathcal{O}(p_i s_i)$. For the sake of simplicity, we can approximate the required matrix-vector multiplication operation for all the DeBut factors as $\mathcal{O}(N \cdot \max_{i=\{1, \cdots, N\}} p_i s_i)$. Finally, by multiplying the number of columns in $X$, we can get the complexity of DeBut as $\mathcal{O}((N \cdot \max_{i=\{1, \cdots, N\}} p_i s_i \cdot H_o W_o)$.

## Appendix II: Alternating Least Squares (ALS) Initialization

In Section 3.2, we introduce how to employ ALS to initialize the DeBut factors. Fig. A1 shows the relative errors of ALS approximation of an FC layer in LeNet and a CONV layer in VGG-16-BN using different chains, respectively. The relative error of ALS is defined as $||F - \hat{F}||_2 / ||F||_2$, where $F$ is the pretrained flattened filter matrix, and $\hat{F}$ is the approximation of $F$ by the ALS initialized DeBut factors. For the FC layer of size $[128, 400]$ in LeNet, four sweeps are enough for the error to converge. Compared with LeNet, the CONV layer of size $[512, 4608]$ in VGG-16-BN needs more sweeps for convergence. That said, ten sweeps are enough for large layers in VGG-16-BN to obtain good initialization.

We set the number of sweeps equal to 5 to initialize small layers in our experiments, namely, all layers in LeNet and CONV1∼3 and FC1 layers in VGG-16-BN. On the other hand, we set the number of sweeps equal to 10 for the large layers in VGG-16-BN and ResNet-50.

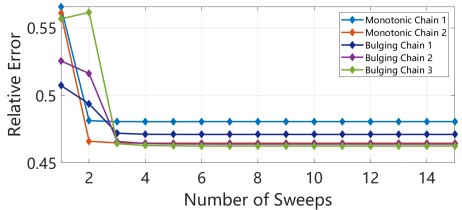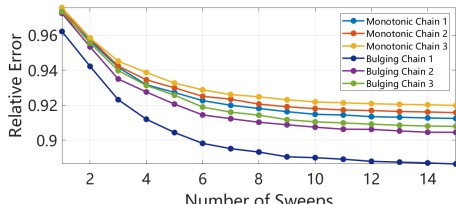

Figure A1: (Left) ALS error plots of DeBut approximation to FC1 layer in the modified LeNet. The five chains are described in Table A1. (Right) ALS error plots of DeBut approximation to CONV12 layer in VGG-16-BN. The six chains are described in Table A5.

## Appendix III: Details of Chains for LeNet

Tables A1 & A2 describe chains for the $[128, 400]$ FC layer and $[16, 72]$ CONV layer in the modified LeNet. Each table contains monotonic and bulging chains. In the last column, the accuracy outside the brackets is obtained without ALS initialization of the DeBut factors, while the number in the brackets shows the performance with ALS initialization.

| Monotonic Chains | LC | ALS Error | Acc. (with ALS) (%) |
|---|---|---|---|
| 1) $128 \xleftarrow{(2,2,64)} 128 \xleftarrow{(2,2,32)} 128 \xleftarrow{(2,2,16)} 128 \xleftarrow{(2,2,8)} 128 \xleftarrow{(8,25,1)} 400$ | 91.75% | 0.9044 | 98.76 (98.56) |
| 2) $128 \xleftarrow{(2,2,64)} 128 \xleftarrow{(2,2,32)} 128 \xleftarrow{(1,2,32)} 256 \xleftarrow{(2,2,16)} 256 \xleftarrow{(16,25,1)} 400$ | 85.00% | 0.8534 | 98.82 (98.74) |
| **Bulging Chains** | **LC** | **ALS Error** | **Acc.(%)** |
| 1) $128 \xleftarrow{(2,4,64)} 256 \xleftarrow{(2,4,32)} 512 \xleftarrow{(4,5,8)} 640 \xleftarrow{(8,5,1)} 400$ | 85.75% | 0.8305 | 98.64 (98.68) |
| 2) $128 \xleftarrow{(2,4,64)} 256 \xleftarrow{(2,4,32)} 512 \xleftarrow{(2,1,16)} 256 \xleftarrow{(16,25,1)} 400$ | 83.50% | 0.8289 | 98.64 (98.69) |
| 3) $128 \xleftarrow{(2,4,64)} 256 \xleftarrow{(1,2,64)} 512 \xleftarrow{(2,2,32)} 512 \xleftarrow{(2,1,16)} 256 \xleftarrow{(16,25,1)} 400$ | 82.50% | 0.8321 | 98.71 (98.86) |

Table A1: Monotonic and bulging DeBut chains to substitute the largest FC layer in the modified LeNet. The layer-wise compression (LC) follows the definition in the main paper.

| Monotonic Chains | LC | ALS Error | Acc. (with ALS) (%) |
|---|---|---|---|
| 1) $16 \xleftarrow{(4,4,4)} 16 \xleftarrow{(2,2,2)} 16 \xleftarrow{(1,3,2)} 48 \xleftarrow{(2,3,1)} 72$ | 75.00% | 0.8030 | 98.87 (99.03) |
| 2) $16 \xleftarrow{(8,8,2)} 16 \xleftarrow{(1,3,2)} 48 \xleftarrow{(2,3,1)} 72$ | 72.22% | 0.8434 | 99.04 (99.05) |
| 3) $16 \xleftarrow{(2,3,8)} 24 \xleftarrow{(1,2,8)} 48 \xleftarrow{(2,2,4)} 48 \xleftarrow{(4,6,1)} 72$ | 58.33% | 0.7248 | 99.00 (99.05) |
| **Bulging Chains** | **LC** | **ALS Error** | **Acc.(%)** |
| 1) $16 \xleftarrow{(2,2,8)} 16 \xleftarrow{(2,6,4)} 48 \xleftarrow{(1,2,4)} 96 \xleftarrow{(4,3,1)} 72$ | 55.56% | 0.6289 | 99.12 (99.03) |
| 2) $16 \xleftarrow{(2,3,8)} 24 \xleftarrow{(1,2,8)} 48 \xleftarrow{(2,4,4)} 96 \xleftarrow{(4,3,1)} 72$ | 50.00% | 0.6595 | 98.95 (99.03) |
| 3) $16 \xleftarrow{(2,6,8)} 48 \xleftarrow{(1,2,8)} 96 \xleftarrow{(2,2,4)} 96 \xleftarrow{(4,3,1)} 72$ | 41.67% | 0.7042 | 99.10 (98.96) |

Table A2: Monotonic and bulging DeBut chains to substitute the largest CONV layer in the modified LeNet. The layer-wise compression (LC) follows the definition in the main paper.

## Appendix IV: Details of Chains for VGG-16-BN

In Table A3, we give three monotonic and three bulging chains for substituting the CONV13 layer in VGG-16-BN.

The sizes of the flattened layers in VGG-16-BN are listed in Table 1. There are ten different sizes: $[64, 27]$ (CONV1), $[64, 576]$ (CONV2), $[128, 576]$ (CONV3), $[128, 1152]$ (CONV4), $[256, 1152]$

(CONV5), $[256, 2304]$ (CONV6-7), $[512, 2304]$ (CONV8), $[512, 4608]$ (CONV9-13), $[512, 512]$ (FC1) and $[512, 10]$ (FC2). Since the number of input channels of CONV1 (3) and the number of output channels of FC2 (10) are small, we do not use DeBut factors to substitue the two layers. For CONV layers of size $[256, 2304]$ and $[512, 4608]$, we select CONV7 and CONV12 as the representatives.

In Table A4, we list monotonic and bulging chains for the flattened layers of size $[64, 576]$, $[128, 576]$, $[128, 1152]$ and $[256, 512]$, and provide the LC and (relative) ALS errors after initialization.

In Table A5, we show monotonic and bulging chains for the flattened layers of size $[256, 2304]$, $[512, 2304]$, $[512, 4608]$. Since the FC layer is of size $[512, 512]$, a square matrix, we employ the regular Butterfly chain. The LC and (relative) ALS errors are shown as well.

| Monotonic Chain(s) | LC | ALS Error | Acc. (with ALS) (%) |
|---|---|---|---|
| 1) $2048 \xleftarrow{(2,2,32)} 2048 \xleftarrow{(2,2,16)} 2048 \xleftarrow{(2,4,8)} 4096 \xleftarrow{(8,9,1)} 4608$ 
 $512 \xleftarrow{(2,4,256)} 1024 \xleftarrow{(2,4,128)} 2048 \xleftarrow{(2,2,64)}$ | 97.31% | 0.9141 | 92.97 (93.90) |
| 2) $2048 \xleftarrow{(2,4,32)} 4096 \xleftarrow{(2,2,16)} 4096 \xleftarrow{(2,2,8)} 4096 \xleftarrow{(8,9,1)} 4608$ 
 $512 \xleftarrow{(2,4,256)} 1024 \xleftarrow{(2,2,128)} 1024 \xleftarrow{(2,4,64)}$ | 97.05% | 0.9307 | 93.09 (93.73) |
| 3) $4096 \xleftarrow{(2,2,32)} 4096 \xleftarrow{(2,2,16)} 4096 \xleftarrow{(2,2,8)} 4096 \xleftarrow{(8,9,1)} 4608$ 
 $512 \xleftarrow{(2,4,256)} 1024 \xleftarrow{(2,4,128)} 2048 \xleftarrow{(2,4,64)}$ | 96.79% | 0.9111 | 93.18 (94.07) |

| Bulging Chain(s) | LC | ALS Error | Acc. (with ALS) (%) |
|---|---|---|---|
| 1) $4096 \xleftarrow{(2,2,32)} 4096 \xleftarrow{(2,4,16)} 8192 \xleftarrow{(4,3,4)} 6144 \xleftarrow{(4,3,1)} 4608$ 
 $512 \xleftarrow{(2,4,256)} 1024 \xleftarrow{(2,4,128)} 2048 \xleftarrow{(2,4,64)}$ | 96.53% | 0.9116 | 93.23 (93.79) |
| 2) $4096 \xleftarrow{(2,4,32)} 8192 \xleftarrow{(2,2,16)} 8192 \xleftarrow{(4,3,4)} 6144 \xleftarrow{(4,3,1)} 4608$ 
 $512 \xleftarrow{(2,4,256)} 1024 \xleftarrow{(2,4,128)} 2048 \xleftarrow{(2,4,64)}$ | 96.18% | 0.9261 | 92.69 (93.86) |
| 3) $4096 \xleftarrow{(2,4,32)} 8192 \xleftarrow{(2,2,16)} 8192 \xleftarrow{(16,9,1)} 4608$ 
 $512 \xleftarrow{(2,4,256)} 1024 \xleftarrow{(2,4,128)} 2048 \xleftarrow{(2,4,64)}$ | 94.88% | 0.9121 | 92.89 (93.93) |

Table A3: DeBut substitution of the last CONV layer in the modified VGG-16-BN.

## Appendix V: Details of Chains for ResNet-50

Tables A6 & A7 describe the chains we use for each CONV layer in the last three blocks except the downsampling layers.

## References

[1] Kaiming He, Xiangyu Zhang, Shaoqing Ren, and Jian Sun. Deep residual learning for image recognition. In *Proceedings of the IEEE conference on computer vision and pattern recognition*, pages 770–778, 2016.

| Layer size | Monotonic Chains | LC | ALS Error |
|---|---|---|---|
| [64, 576] (CONV2) | 1) $64 \xleftarrow{(8,9,8)} 72 \xleftarrow{(2,4,4)} 144 \xleftarrow{(1,2,4)} 288 \xleftarrow{(4,8,1)} 576$ | 90.62% | 0.9480 |
| | 2) $64 \xleftarrow{(8,16,8)} 128 \xleftarrow{(1,2,8)} 256 \xleftarrow{(2,3,4)} 384 \xleftarrow{(4,6,1)} 576$ | 88.19% | 0.9478 |
| | 3) $64 \xleftarrow{(4,8,16)} 128 \xleftarrow{(1,2,16)} 256 \xleftarrow{(2,3,8)} 384 \xleftarrow{(8,12,1)} 576$ | 83.33% | 0.8808 |

| | Bulging Chains | LC | ALS Error |
|---|---|---|---|
| [64, 576] (CONV2) | 1) $64 \xleftarrow{(4,8,16)} 128 \xleftarrow{(2,4,8)} 256 \xleftarrow{(2,3,4)} 384 \xleftarrow{(1,2,4)} 768 \xleftarrow{(4,3,1)} 576$ | 86.81% | 0.9182 |
| | 2) $64 \xleftarrow{(2,4,32)} 128 \xleftarrow{(2,4,16)} 256 \xleftarrow{(2,3,8)} 384 \xleftarrow{(2,4,4)} 768 \xleftarrow{(4,3,1)} 576$ | 85.42% | 0.8369 |
| | 3) $64 \xleftarrow{(2,4,32)} 128 \xleftarrow{(2,4,16)} 256 \xleftarrow{(2,3,8)} 384 \xleftarrow{(1,2,8)} 768 \xleftarrow{(8,6,1)} 576$ | 81.25% | 0.8295 |

| Layer size | Monotonic Chains | LC | ALS Error |
|---|---|---|---|
| [128, 576] (CONV3) | 1) $128 \xleftarrow{(4,8,32)} 256 \xleftarrow{(2,2,16)} 256 \xleftarrow{(4,6,4)} 384 \xleftarrow{(4,6,1)} 576$ | 92.71% | 0.9531 |
| | 2) $128 \xleftarrow{(2,4,64)} 256 \xleftarrow{(4,4,16)} 256 \xleftarrow{(4,6,4)} 384 \xleftarrow{(4,6,1)} 576$ | 92.71% | 0.9135 |
| | 3) $128 \xleftarrow{(8,16,16)} 256 \xleftarrow{(2,2,8)} 256 \xleftarrow{(2,3,4)} 384 \xleftarrow{(4,6,1)} 576$ | 92.36% | 0.9720 |

| | Bulging Chains | LC | ALS Error |
|---|---|---|---|
| [128, 576] (CONV3) | 1) $128 \xleftarrow{(4,4,32)} 128 \xleftarrow{(4,8,8)} 256 \xleftarrow{(2,3,4)} 384 \xleftarrow{(1,2,4)} 768 \xleftarrow{(4,3,1)} 576$ | 92.71% | 0.9287 |
| | 2) $128 \xleftarrow{(4,4,32)} 128 \xleftarrow{(2,4,16)} 256 \xleftarrow{(2,3,8)} 384 \xleftarrow{(2,4,4)} 768 \xleftarrow{(4,3,1)} 576$ | 92.37% | 0.9274 |
| | 3) $128 \xleftarrow{(2,2,64)} 128 \xleftarrow{(2,4,32)} 256 \xleftarrow{(2,3,16)} 384 \xleftarrow{(2,4,8)} 768 \xleftarrow{(8,6,1)} 576$ | 89.59% | 0.8926 |

| Layer size | Monotonic Chains | LC | ALS Error |
|---|---|---|---|
| [128, 1152] (CONV4) | 1) $128 \xleftarrow{(4,8,32)} 256 \xleftarrow{(2,4,16)} 512 \xleftarrow{(2,4,8)} 1024 \xleftarrow{(8,9,1)} 1152$ | 90.97% | 0.9424 |
| | 2) $128 \xleftarrow{(4,8,32)} 256 \xleftarrow{(4,8,8)} 512 \xleftarrow{(1,2,8)} 1024 \xleftarrow{(8,9,1)} 1152$ | 90.97% | 0.9413 |
| | 3) $128 \xleftarrow{(2,4,64)} 256 \xleftarrow{(8,16,8)} 512 \xleftarrow{(1,2,8)} 1024 \xleftarrow{(8,9,1)} 1152$ | 89.93% | 0.9135 |

| | Bulging Chains | LC | ALS Error |
|---|---|---|---|
| [128, 1152] (CONV4) | 1) $128 \xleftarrow{(2,4,64)} 256 \xleftarrow{(2,4,32)} 512 \xleftarrow{(2,3,16)} 768 \xleftarrow{(2,4,8)} 1536 \xleftarrow{(8,6,1)} 1152$ | 89.58% | 0.9195 |
| | 2) $128 \xleftarrow{(2,4,64)} 256 \xleftarrow{(1,2,64)} 512 \xleftarrow{(4,6,16)} 768 \xleftarrow{(2,4,8)} 1536 \xleftarrow{(8,6,1)} 1152$ | 88.89% | 0.9132 |
| | 3) $128 \xleftarrow{(1,2,128)} 256 \xleftarrow{(2,4,64)} 512 \xleftarrow{(4,6,16)} 512 \xleftarrow{(2,4,8)} 256 \xleftarrow{(18,6,1)} 400$ | 88.72% | 0.8983 |

| Layer size | Monotonic Chains | LC | ALS Error |
|---|---|---|---|
| [256, 1152] (CONV5) | 1) $256 \xleftarrow{(8,16,32)} 512 \xleftarrow{(2,2,16)} 512 \xleftarrow{(2,4,8)} 1024 \xleftarrow{(8,9,1)} 1152$ | 91.44% | 0.9780 |
| | 2) $256 \xleftarrow{(4,8,64)} 512 \xleftarrow{(2,2,32)} 512 \xleftarrow{(4,8,8)} 1024 \xleftarrow{(8,9,1)} 1152$ | 91.44% | 0.9681 |
| | 3) $256 \xleftarrow{(4,8,64)} 512 \xleftarrow{(8,8,8)} 512 \xleftarrow{(1,2,8)} 1024 \xleftarrow{(8,9,1)} 1152$ | 91.44% | 0.9662 |

| | Bulging Chains | LC | ALS Error |
|---|---|---|---|
| [256, 1152] (CONV5) | 1) $256 \xleftarrow{(2,4,128)} 512 \xleftarrow{(4,6,32)} 768 \xleftarrow{(8,16,4)} 1536 \xleftarrow{(4,3,1)} 1152$ | 94.62% | 0.9685 |
| | 2) $256 \xleftarrow{(2,4,128)} 512 \xleftarrow{(4,6,32)} 768 \xleftarrow{(8,16,4)} 1536 \xleftarrow{(4,3,1)} 1152$ | 92.88% | 0.9403 |
| | 3) $256 \xleftarrow{(2,4,128)} 512 \xleftarrow{(16,24,8)} 768 \xleftarrow{(2,4,4)} 1536 \xleftarrow{(4,3,1)} 1152$ | 92.88% | 0.9401 |

Table A4: DeBut chains for layers with flattened weight matrices of sizes $[64, 576]$, $[128, 576]$, $[128, 1152]$, and $[256, 1152]$.

| Layer size | Monotonic Chains | | LC | ALS Error |
|---|---|---|---|---|
| [256, 2304]
(CONV7) | 1) $256 \xleftarrow{(4,8,64)} 512 \xleftarrow{(2,4,32)} 1024 \xleftarrow{(4,8,8)} 2048 \xleftarrow{(8,9,1)} 2304$ | | 94.79% | 0.9596 |
| | 2) $256 \xleftarrow{(2,4,128)} 512 \xleftarrow{(4,8,32)} 1024 \xleftarrow{(4,8,8)} 2048 \xleftarrow{(8,9,1)} 2304$ | | 94.62% | 0.9506 |
| | 3) $256 \xleftarrow{(2,4,128)} 512 \xleftarrow{(2,4,64)} 1024 \xleftarrow{(8,16,8)} 2048 \xleftarrow{(8,9,1)} 2304$ | | 93.58% | 0.9420 |

| Layer size | Bulging Chains | | LC | ALS Error |
|---|---|---|---|---|
| [256, 2304]
(CONV7) | 1) $256 \xleftarrow{(2,4,128)} 512 \xleftarrow{(1,2,128)} 1024 \xleftarrow{(4,8,32)} 2048 \xleftarrow{(4,6,8)} 3072 \xleftarrow{(8,6,1)} 2304$ | | 93.06% | 0.9405 |
| | 2) $256 \xleftarrow{(2,4,128)} 512 \xleftarrow{(1,2,128)} 1024 \xleftarrow{(8,16,16)} 2048 \xleftarrow{(2,3,8)} 3072 \xleftarrow{(8,6,1)} 2304$ | | 92.71% | 0.9391 |
| | 3) $256 \xleftarrow{(2,4,128)} 512 \xleftarrow{(2,4,64)} 1024 \xleftarrow{(2,4,32)} 2048 \xleftarrow{(2,3,16)} 3072 \xleftarrow{(16,12,1)} 2304$ | | 91.49% | 0.9357 |

| Layer size | Monotonic Chains | | LC | ALS Error |
|---|---|---|---|---|
| [512, 2304]
(CONV8) | 1) $512 \xleftarrow{(8,16,64)} 1024 \xleftarrow{(4,4,16)} 1024 \xleftarrow{(2,4,8)} 2048 \xleftarrow{(8,9,1)} 2304$ | | 97.05% | 0.9854 |
| | 2) $\xleftarrow{(2,4,64)} 2048 \xleftarrow{(2,2,32)} 2048 \xleftarrow{(2,2,16)} 2048 \xleftarrow{(2,2,8)} 2048 \xleftarrow{(8,9,1)} 2304$
$512 \xleftarrow{(2,2,256)} 512 \xleftarrow{(2,4,128)} 1024$ | | 96.79% | 0.9654 |
| | 3) $512 \xleftarrow{(4,8,128)} 1024 \xleftarrow{(4,8,32)} 1024 \xleftarrow{(4,4,8)} 2048 \xleftarrow{(8,9,1)} 2304$ | | 96.70% | 0.9749 |

| Layer size | Bulging Chains | | LC | ALS Error |
|---|---|---|---|---|
| [512, 2304]
(CONV8) | 1) $512 \xleftarrow{(4,8,128)} 1024 \xleftarrow{(2,4,64)} 2048 \xleftarrow{(2,2,32)} 2048 \xleftarrow{(4,6,1)} 3072 \xleftarrow{(8,6,1)} 2304$ | | 96.35% | 0.9755 |
| | 2) $512 \xleftarrow{(2,4,256)} 1024 \xleftarrow{(2,4,128)} 2048 \xleftarrow{(4,4,32)} 2048 \xleftarrow{(4,6,8)} 3072 \xleftarrow{(8,6,1)} 2304$ | | 96.18% | 0.9578 |
| | 3) $512 \xleftarrow{(2,4,256)} 1024 \xleftarrow{(2,4,128)} 2048 \xleftarrow{(4,4,32)} 2048 \xleftarrow{(2,3,16)} 3072 \xleftarrow{(16,12,1)} 2304$ | | 95.14% | 0.9509 |

| Layer size | Monotonic Chains | | LC | ALS Error |
|---|---|---|---|---|
| (CONV12) | 1) $\xleftarrow{(2,2,64)} 2048 \xleftarrow{(2,2,32)} 2048 \xleftarrow{(2,2,16)} 2048 \xleftarrow{(2,4,8)} 4096 \xleftarrow{(8,9,1)} 4608$
$512 \xleftarrow{(2,4,256)} 1024 \xleftarrow{(2,4,128)} 2048$ | | 97.31% | 0.9199 |
| | 2) $\xleftarrow{(2,4,64)} 2048 \xleftarrow{(2,4,32)} 4096 \xleftarrow{(2,2,16)} 4096 \xleftarrow{(2,2,8)} 4096 \xleftarrow{(8,9,1)} 4608$
$512 \xleftarrow{(2,4,256)} 1024 \xleftarrow{(2,2,128)} 1024$ | | 97.05% | 0.9158 |
| | 3) $\xleftarrow{(2,4,64)} 4096 \xleftarrow{(2,2,32)} 4096 \xleftarrow{(2,2,16)} 4096 \xleftarrow{(2,2,8)} 4096 \xleftarrow{(8,9,1)} 4608$
$512 \xleftarrow{(2,4,256)} 1024 \xleftarrow{(2,4,128)} 2048$ | | 96.79% | 0.9124 |

| Layer size | Bulging Chains | | LC | ALS Error |
|---|---|---|---|---|
| [512, 4608]

(CONV12) | 1) $\xleftarrow{(2,4,64)} 4096 \xleftarrow{(2,2,32)} 4096 \xleftarrow{(2,4,16)} 8192 \xleftarrow{(4,3,4)} 6144 \xleftarrow{(4,3,1)} 4096$
$512 \xleftarrow{(2,4,256)} 1024 \xleftarrow{(2,4,128)} 2048$ | | 96.53% | 0.9075 |
| | 2) $\xleftarrow{(2,4,64)} 4096 \xleftarrow{(2,4,32)} 8192 \xleftarrow{(2,2,16)} 8192 \xleftarrow{(4,3,4)} 6144 \xleftarrow{(4,3,1)} 4096$
$512 \xleftarrow{(2,4,256)} 1024 \xleftarrow{(2,4,128)} 2048$ | | 96.18% | 0.9045 |
| | 3) $\xleftarrow{(2,4,128)} 2048 \xleftarrow{(2,4,64)} 4096 \xleftarrow{(2,4,32)} 8192 \xleftarrow{(2,2,16)} 8192 \xleftarrow{(16,9,1)} 4608$
$512 \xleftarrow{(2,4,256)} 1024$ | | 94.88% | 0.8864 |

| Layer size | Regular Chains | | LC | ALS Error |
|---|---|---|---|---|
| [512, 512]
(FC1) | 1) $\xleftarrow{(2,2,8)} 512 \xleftarrow{(2,2,16)} 512 \xleftarrow{(2,2,32)} 512 \xleftarrow{(2,2,64)} 512 \xleftarrow{(2,2,128)} 512 \xleftarrow{(2,2,256)} 512$
$512 \xleftarrow{(2,2,1)} 512 \xleftarrow{(2,2,2)} 512 \xleftarrow{(2,2,4)} 512$ | | 96.48% | 0.9010 |

Table A5: DeBut chains for layers with flattened weight matrices of sizes $[256, 2304]$, $[512, 2304]$, $[512, 4608]$, and $[512, 512]$.

| Layer | Chains | LC | ALS Error |
|---|---|---|---|
| CONV5_1 | $1024 \xleftarrow{(2,4,16)} 2048 \xleftarrow{(2,2,8)} 2048 \xleftarrow{(2,2,4)} 2048 \xleftarrow{(4,2,1)} 1024$ 
 $512 \xleftarrow{(2,2,256)} 512 \xleftarrow{(2,4,128)} 1024 \xleftarrow{(2,2,64)} 1024 \xleftarrow{(2,2,32)}$ | 95.51% | 0.9630 |
| CONV5_2 | $4096 \xleftarrow{(2,4,32)} 8192 \xleftarrow{(2,2,16)} 8192 \xleftarrow{(16,9,1)} 4608$ 
 $512 \xleftarrow{(2,4,256)} 1024 \xleftarrow{(2,4,128)} 2048 \xleftarrow{(2,4,64)}$ | 94.88% | 0.9599 |
| CONV5_3 | $1024 \xleftarrow{(2,2,32)} 1024 \xleftarrow{(2,2,16)} 1024 \xleftarrow{(2,2,8)} 1024 \xleftarrow{(2,2,4)} 1024 \xleftarrow{(4,2,1)} 512$ 
 $2048 \xleftarrow{(2,2,1024)} 2048 \xleftarrow{(2,2,512)} 2048 \xleftarrow{(4,2,128)} 1024 \xleftarrow{(2,2,64)}$ | 97.66% | 0.9788 |
| CONV5_4 | $8192 \xleftarrow{(4,4,16)} 8192 \xleftarrow{(4,4,4)} 8192 \xleftarrow{(4,1,1)} 2048$ 
 $512 \xleftarrow{(2,8,256)} 2048 \xleftarrow{(4,16,64)}$ | 89.45% | 0.9164 |
| CONV5_5 | $4096 \xleftarrow{(2,4,32)} 8192 \xleftarrow{(2,2,16)} 8192 \xleftarrow{(16,9,1)} 4608$ 
 $512 \xleftarrow{(2,4,256)} 1024 \xleftarrow{(2,4,128)} 2048 \xleftarrow{(2,4,64)}$ | 94.88% | 0.9567 |
| CONV5_6 | $1024 \xleftarrow{(2,2,32)} 1024 \xleftarrow{(2,2,16)} 1024 \xleftarrow{(2,2,8)} 1024 \xleftarrow{(2,2,4)} 1024 \xleftarrow{(4,2,1)} 512$ 
 $2048 \xleftarrow{(2,2,1024)} 2048 \xleftarrow{(2,2,512)} 2048 \xleftarrow{(4,2,128)} 1024 \xleftarrow{(2,2,64)}$ | 97.66% | 0.9791 |
| CONV5_7 | $8192 \xleftarrow{(4,4,16)} 8192 \xleftarrow{(4,4,4)} 8192 \xleftarrow{(4,1,1)} 2048$ 
 $512 \xleftarrow{(2,8,256)} 2048 \xleftarrow{(4,16,64)}$ | 89.45% | 0.9189 |
| CONV5_8 | $4096 \xleftarrow{(2,4,32)} 8192 \xleftarrow{(2,2,16)} 8192 \xleftarrow{(16,9,1)} 4608$ 
 $512 \xleftarrow{(2,4,256)} 1024 \xleftarrow{(2,4,128)} 2048 \xleftarrow{(2,4,64)}$ | 94.88% | 0.9544 |
| CONV5_9 | $1024 \xleftarrow{(2,2,32)} 1024 \xleftarrow{(2,2,16)} 1024 \xleftarrow{(2,2,8)} 1024 \xleftarrow{(2,2,4)} 1024 \xleftarrow{(4,2,1)} 512$ 
 $2048 \xleftarrow{(2,2,1024)} 2048 \xleftarrow{(2,2,512)} 2048 \xleftarrow{(4,2,128)} 1024 \xleftarrow{(2,2,64)}$ | 97.66% | 0.9778 |

Table A6: The bulging chains for DeBut-bulging. CONV5_1 to CONV5_9 are convolution layers from the last three blocks denoted in [1].

| Layer | Chains | LC | ALS Error |
|---|---|---|---|
| CONV5_1 | $512 \xleftarrow{(2,4,256)} 1024 \xleftarrow{(4,4,64)} 1024 \xleftarrow{(4,4,16)} 1024 \xleftarrow{(4,4,4)} 1024 \xleftarrow{(4,4,1)} 1024$ | 96.48% | 0.9640 |
| CONV5_2 | $4096 \xleftarrow{(2,2,32)} 4096 \xleftarrow{(2,2,16)} 4096 \xleftarrow{(2,2,8)} 4096 \xleftarrow{(8,9,1)} 4608$ 
 $512 \xleftarrow{(2,4,256)} 1024 \xleftarrow{(2,4,128)} 2048 \xleftarrow{(2,4,64)}$ | 96.79% | 0.9730 |
| CONV5_3 | $1024 \xleftarrow{(2,2,32)} 1024 \xleftarrow{(2,2,16)} 1024 \xleftarrow{(2,2,8)} 1024 \xleftarrow{(2,2,4)} 1024 \xleftarrow{(4,2,1)} 512$ 
 $2048 \xleftarrow{(2,2,1024)} 2048 \xleftarrow{(2,2,512)} 2048 \xleftarrow{(4,2,128)} 1024 \xleftarrow{(2,2,64)}$ | 97.66% | 0.9788 |
| CONV5_4 | $1024 \xleftarrow{(2,4,16)} 2048 \xleftarrow{(4,4,4)} 2048 \xleftarrow{(4,4,1)} 2048$ 
 $512 \xleftarrow{(2,2,256)} 512 \xleftarrow{(2,4,128)} 1024 \xleftarrow{(4,4,32)}$ | 97.36% | 0.9705 |
| CONV5_5 | $4096 \xleftarrow{(2,2,32)} 4096 \xleftarrow{(2,2,16)} 4096 \xleftarrow{(2,2,8)} 4096 \xleftarrow{(8,9,1)} 4608$ 
 $512 \xleftarrow{(2,4,256)} 1024 \xleftarrow{(2,4,128)} 2048 \xleftarrow{(2,4,64)}$ | 96.79% | 0.9699 |
| CONV5_6 | $1024 \xleftarrow{(2,2,32)} 1024 \xleftarrow{(2,2,16)} 1024 \xleftarrow{(2,2,8)} 1024 \xleftarrow{(2,2,4)} 1024 \xleftarrow{(4,2,1)} 512$ 
 $2048 \xleftarrow{(2,2,1024)} 2048 \xleftarrow{(2,2,512)} 2048 \xleftarrow{(4,2,128)} 1024 \xleftarrow{(2,2,64)}$ | 97.66% | 0.9791 |
| CONV5_7 | $1024 \xleftarrow{(2,4,16)} 2048 \xleftarrow{(4,4,4)} 2048 \xleftarrow{(4,4,1)} 2048$ 
 $512 \xleftarrow{(2,2,256)} 512 \xleftarrow{(2,4,128)} 1024 \xleftarrow{(4,4,32)}$ | 97.36% | 0.9716 |
| CONV5_8 | $4096 \xleftarrow{(2,2,32)} 4096 \xleftarrow{(2,2,16)} 4096 \xleftarrow{(2,2,8)} 4096 \xleftarrow{(8,9,1)} 4608$ 
 $512 \xleftarrow{(2,4,256)} 1024 \xleftarrow{(2,4,128)} 2048 \xleftarrow{(2,4,64)}$ | 96.79% | 0.9669 |
| CONV5_9 | $1024 \xleftarrow{(2,2,32)} 1024 \xleftarrow{(2,2,16)} 1024 \xleftarrow{(2,2,8)} 1024 \xleftarrow{(2,2,4)} 1024 \xleftarrow{(4,2,1)} 512$ 
 $2048 \xleftarrow{(2,2,1024)} 2048 \xleftarrow{(2,2,512)} 2048 \xleftarrow{(4,2,128)} 1024 \xleftarrow{(2,2,64)}$ | 97.66% | 0.9778 |

Table A7: The monotonic chains for DeBut-mono. CONV5_1 to CONV5_9 are convolution layers from the last three blocks denoted in [1].