# OpenReview forum: "Deformable Butterfly: A Highly Structured and Sparse Linear Transform"
_NeurIPS.cc/2021/Conference — NeurIPS 2021 Poster_

### Official Review · Reviewer_8jnC · 2021-07-14

**Rating:** 4
**Confidence:** 4

**Summary:**

The authors propose “deformable butterfly”, a variant of the butterfly transform that does not require power-of-two size butterfly factor matrices. The authors show how to adapt pre-trained models to use DeBut in place of convolution and fully-connected layers and also show experiments training models with DeBut chains from scratch. The authors provide experiments on a range of CNN architectures and datasets and compare to existing transforms like Adaptive Fastfood and Butterfly.

**Ethical Concerns:**

I did not identify ethical issues with this paper.

**Limitations And Societal Impact:**

I did not identify potential negative societal impacts in this work.

**Main Review:**

The paper is well written and organized. I found the work very interesting, but my main concerns are novelty and unique contribution over existing work. As I understand it, the primary contribution in methodology is that DeBut breaks the power-of-two (POT) constraint required in standard Butterfly matrices. This is certainly practically useful, but I don’t think this change is novel enough to merit publication for this contribution alone. Additional contributions like empirical gains over Butterfly matrices would’ve helped, but the advantage over Butterfly matrices seems to be marginal. Both techniques still do not appear to be simple substitutes for existing layers without accuracy loss. For example, 0.65% accuracy drop (almost 2x increase in error) training from scratch when replacing 3 out of 5 layers in LeNet and 1.5% decrease in top-1 accuracy using ALS with ResNet-50 with DeBut applied to only 9 layers in the model.

Comments
1. Some of the language in this work is very sensational. For example, “renders a new paradigm for network compression”.
2. In practice, convolution is rarely implemented with im2col + GEMM because of the memory bandwidth cost and memory footprint of the im2col. Modern high-performance convolutions fuse the im2col + GEMM into a single kernel with clever indexing (called an “implicit-GEMM” convolution, as opposed to “explicit-GEMM” where im2col is used).
3. The authors should include the baseline accuracy in their result tables to make it easier for the reader.
4. I don’t see how DeBut “unif[ies] and homogenize[s]” convolution and FC layers more than they already are. DeBut using an im2col transform and then approximating the GEMM portion of the convolution is just exploiting what was already known.
5. The statement “where other fast linear transform schemes fail” is an exaggeration. The empirical gains over Butterfly appear to be limited and Adaptive FastFood produces much better theoretical efficiency (although it appears to be expensive in practice).
6. I’d like to see more details on the implementations benchmarked in Table 6. Given all techniques listed are not mainstream I doubt the implementations are highly optimized and I’d like to understand how much this contributes to the performance discrepancies. For example, I’m not sure why DeBut would be much faster than Butterfly given they are essentially the same computation.

**Time Spent Reviewing:**

2hrs

---

> ### Author Response · Authors · 2021-08-10
> **Thanks for your insightful feedback.**
>
> Thanks for your comments. Besides breaking the PoT constraint, we also devise different geometries for the chains (that's why we call it deformable) and their merits in largely reducing the number of parameters. In fact, other model compression methods like low-rank decomposition also trade accuracy for the number of parameters, and the drop in accuracy in DeBut nets are on the very mild side given the compression ratio (Table shown in response to Reviewer aaRU, Q3).
>
> - Q1. Some of the language in this work is very sensational. For example, “renders a new paradigm for network compression”.
> - A1. We will revise the wording to be less sensational, e.g., "opens up a new room for performing network compression".
>
> - Q2. In practice, convolution is rarely implemented with im2col + GEMM because of the memory bandwidth cost and memory footprint of the im2col. Modern high-performance convolutions fuse the im2col + GEMM into a single kernel with clever indexing (called an “implicit-GEMM” convolution, as opposed to “explicit-GEMM” where im2col is used).
> - A2. We want to highlight that cuFFT (a highly specialized and optimized FFT) is about 5 times faster than WHT even though the latter should have a lower complexity. The reason is that specialized algorithmic and hardware-level optimization can be done to speedup FFT due to its widespread use. The goal of our work is to introduce the DeBut framework, whose optimization would follow in our future work. In fact, between paper submission and now, we have already worked out variants of DeBut (e.g., it still performs well in the heavily quantized BNNs, namely, more than halving the number of parameters with only 0.3% drop in Top-1 accuracy).
>
> |  Model     |    Method     | Bit-width(Weight/Activation) | Accuracy(%) | #params |
> | ------------- | ----------------- | -------------------- | ---------------- | ---------- |
> | ResNet-18 | Full precision  |    32/32           |    93.0        | 11.28M  |
> | ResNet-18 |   RBNN[1]      |     1/1            |     92.2       | 11.28M  |
> | ResNet-18 |   DeBut         |     1/1             |     91.9       | 4.56M   |
>
> [1] Lin, Mingbao, et al. "Rotated binary neural network."
>
> - Q3. The authors should include the baseline accuracy in their result tables to make it easier for the reader.
> - A3. Thanks for the editorial suggestion. We will revise the accepted paper listing the baseline in the tables.
>
> - Q4. I don’t see how DeBut “unif[ies] and homogenize[s]” convolution and FC layers more than they already are.
> - A4. Yes and No. DeBut DOES NOT further unify and homogenize FC and CONV in the sense that in the im2col setting, both conv and FC are linear transforms via matrix multiplication, and so is DeBut. But DeBut DOES further unify and homogenize FC and CONV in the sense that DeBut is doing this linear mapping in its unique butterfly-like attribute with reduced complexity than dense matrix multiplication. This can be analogized to the 2014 work (Striving for Simplicity: The All Convolutional Net) that homogenizes the network into CNN layers. Nonetheless, to avoid misunderstanding, we will change the wordings accordingly to emphasize DeBut stands for a structured matrix decomposition that can substitute (esp. large) CNN and/or FC layers with much reduced number of parameters while not harming the accuracy much.
>
> - Q5. The statement “where other fast linear transform schemes fail” is an exaggeration. The empirical gains over Butterfly appear to be limited and Adaptive FastFood produces much better theoretical efficiency (although it appears to be expensive in practice).
> - A5. Yes, as mentioned above, cuFFT is around 5 times faster than standard codes of WHT, though the latter should have a lower complexity. We agree that once DeBut is found to be meaningful and useful, its fully optimized (algorithmic & hardware-level) version would run at least as efficient as competing algorithms.
>
> - A6. I’d like to see more details on the implementations benchmarked in Table 6. Given all techniques listed are not mainstream I doubt the implementations are highly optimized and I’d like to understand how much this contributes to the performance discrepancies.
> - A6. The results reported in Table 6 of Butterfly are our implementation using their officially released codes, Adaptive FastFood is modified by the official Fastfood codes on sklearn. For DeBut, the chains for the corresponding layers are the same as described in Table 3. All experiments are conducted on an NVIDIA GeForce GTX1080 Ti Graphics Card with 11GB frame buffer, and our codes are provided in the Supplementary.

---

> > ### Comment · Reviewer_8jnC · 2021-08-13
> > **Reviewer Response**
> >
> > Thank you to the authors for their responses.
> >
> > I remain unconvinced regarding the contributions of this paper. As I stated previously, the changes over the standard Butterfly technique aren’t particularly large and the empirical results don’t seem to show much of an advantage over the baselines.
> >
> > I’m skeptical that the comparisons of training and inference times in Table 6 are accurate given the fact that Adaptive FastFood is running in sklearn on CPU (sklearn does not provide GPU support). Given the butterfly methods are using a GPU I think the comparison is misleading, especially given Adaptive FastFood shows better theoretical compression for a given accuracy in Table 5.

---

> > > ### Author Response · Authors · 2021-08-17
> > > **DeBut is indeed theoretically and practically rich**
> > >
> > > Thanks for your comments. It took us some time to prepare more experimental results with respect to your concern, and here is our further response.
> > >
> > > First, we believe the idea of generalizing from powers-of-two (PoT) standard Butterfly to the now “deformable” chain geometry with arbitrarily sized dense blocks (on the rightmost butterfly factor) is first-of-its-kind, innovative, and non-trivial (cf. more experimental results below). DeBut is obviously in contrast to the existing FC substitutes via Adaptive Fastfood, Butterfly, or K-matrix that must conform to PoT square matrices. Using DeBut there can be no wastage in the “dead parts” seen in the former mentioned approaches. Given that non-PoT Butterfly is non-existent in the literature, we believe such a new dimension of thinking is the true value of this work. In fact, for CNN we have seen a series of works on deformable convolutional networks, whose primitive idea came out around 2017 and then underwent various extensions and improvements. If the idea was nullified at its infancy, then there would not have been a series of successful sequel works. More proof on the practicality of DeBut can be seen below.
> > >
> > > In short, DeBut thinks out of the box and provides a new dimension of thinking. More importantly, there is a practical implication to it, namely, the deformable dense blocks serve the purpose of tuning the receptive fields like those in a CNN. This is especially important when we use DeBut to substitute CNN layers with, say, a 3x3 kernel (a 9x1 vector results from the im2col). “Matching” this im2col input with a filter matrix F approximated by a standard butterfly of 2x2 dense blocks cannot produce the accuracy obtained via DeBut with 8*9 dense blocks. We list our experimental results on CIFAR10 with the pretrained model below:
> > >
> > > |     Model      | Rightmost block | #params | Accuracy(%) |
> > > | ---------------------- | -------------------- | ----------- | ---------------- |
> > > | VGG16(Baseline) |     N/A      | 14.99M  |   93.96    |
> > > | VGG16         |     2*2      | 3.58M  |   93.81   |
> > > | VGG16         |     8*9      | 3.57M  |   94.19    |
> > >
> > > The results show even higher accuracy resulting from the DeBut replacement! This alone bears out the importance and also merits of lifting the PoT constraint.
> > >
> > > Moreover, as listed in our earlier reply, DeBut can be applied to extremely quantized binary neural networks. This by itself is an amazing demonstration, and such adaptivity is not possible with Adaptive Fastfood since its diagonal matrices must carry full-precision numbers to maintain accuracy. This again already demonstrates the niche of DeBut over Adaptive Fastfood.
> > >
> > > Finally, we did not mean to jeopardize Adaptive Fastfood by using sklearn. The authors of Adaptive Fastfood did not provide any official codes, and the one we used in our experiments is the fastest repo we could find. That said, we have also tried to code our own Adaptive Fastfood using GPU resources, but it is still too slow to finish the whole training process when we compress more convolution layers. Suppose with a 3x3 kernel, 512-channel input, and 512-channel output, the filter matrix F (cf. Fig. 2) measures [512,4608]. If we are to substitute it using Adaptive Fastfood or standard Butterfly, then the PoT constraint dictates the use of [8192, 8192] matrices (namely, for the Hadamard & Permutation matrices in Fastfood and butterfly factors in standard butterflies) which will lead to slow runtimes with much wastage in the dead (unused) parts of the matrices.  It is worth noting that Fastfood does not provide the solution to deal with the non-square matrix, whose width is larger than the height, and we modified it by employing the zero-padding strategy as in the standard Butterfly. Based on this example, we tested the Fast Hadamard Transform taking an 8192*1 vector as the input using sklearn on CPU, PyTorch on CPU, and GPU, respectively.
> > >
> > >    | Method         | Input size | Average time |
> > >    | -------------- | ---------- | ------------ |
> > >    | Sklearn        | [8192,1]   | 0.0001s      |
> > >    | Pytorch on CPU | [8192,1]   | 6.6058s      |
> > >    | Pytorch on GPU | [8192,1]   | 13.4392s     |
> > >
> > > Results show the sklearn codes (actually written in C) are reasonably fast, but still inefficient since Adaptive Fastfood requires Fast Hadamard Transform in both forward and backward passes.
> > >
> > > Regarding “the changes over the standard Butterfly technique aren’t particularly large”, we argue that since DeBut allows deformed geometry of the chains, there comes a myriad of possible chains whose formation and auto-generation are actually rather non-trivial. For this we have coded a script, per request of other reviewers too, to automatically produce the mono and bulging chains which we have placed in the repo: https://anonymous.4open.science/r/Rebuttal-773-6DF6
> > >
> > > Once again, we want to bring back the focus to the innovation in DeBut which is the first work, as far as we know, that introduces a non-PoT butterfly, whose unique and significant advantages have been further confirmed by the experiments shown in this response. Obviously, DeBut provides an excellent alternative, if not a replacement, of Adaptive Fastfood and/or K-matrix, and its novelty is definitely worth promoting, as is echoed by all other reviewers.
> > >
> > > We do thank the reviewer for the very useful comments that have pushed us to investigate deeper into the potential of DeBut. All your comments will be handled with uncompromising care when we prepare the final version of this work, and we sincerely hope for your favorable consideration of our work.

---

### Official Review · Reviewer_aaRU · 2021-07-15

**Rating:** 7
**Confidence:** 3

**Summary:**

The paper present a generalization to butterfly matrices used for FC layer compression that allows flexibility in the dimensionality of the map. This removes the power-of-two limitation. Using that, the authors compress not only FC layers but also conv layers. In addition, the DeBut chain can be made flexible and serve as an addition design parameter. The results confirm that expressivity remains high under considerable compression ratios with better inference rates than competing approaches.

**Limitations And Societal Impact:**

limitations are not discussed in depth

**Main Review:**

Strengths:
* I wish to compliment the authors on a very well written paper. It was clear and pleasant to follow.
* The proposed method is elegant and practical. The experimental section shown convincing results on standard benchmarks.
* The hierarchical structure imposed by the DeBut chain is interesting to investigate, and the different chains that can be formed allow a nice flexibility to the model design with possible trade-offs between expressivity and model size.

Weaknesses:
* I would like to see a more principled comparison of different chains and their effect on performance.
* when applied to convolutional layers the hierarchy effectively imposes a spatial structure to grouping neighboring pixels. Could you show those patterns? would it make sense to shuffle the column vectors resulting from im2col to control these patterns?
* In the context of compression, many other methods exist but they're not shown in the paper.

**Time Spent Reviewing:**

2

---

> ### Author Response · Authors · 2021-08-10
> **We sincerely appreciate your positive comments on DeBut**
>
> We sincerely appreciate your positive comments on DeBut which we very much like to introduce to the community. Indeed, we have obtained further interesting results between paper submission and now, such as the BNN variant of DeBut. The hierarchical structure in DeBut also describes an interesting information flow mechanism which we would like to elaborate in a future work too.
>
> - Q1. I would like to see a more principled comparison of different chains and their effect on performance.
> - A1. We have a more detailed description and evaluation standards about the chain generation process and provide a demo to generate the chains under various requirements. Please check the anonymous Github: https://anonymous.4open.science/r/Rebuttal-773-6DF6, which contains a .ipynb file to show the core codes and the output results.
> In summary, we now evaluate the chains from three aspects: 1) chain type: monotonic or bulging, 2) shrinking speed: fast or slow, 3) compression rate. According to our results, the bulging chain is the first choice if higher accuracy is expected, while monotonic chains are preferred when a more robust model with much fewer parameters is required. We will sort out a Table to show a more principled chain comparison in the accepted paper.
>
> - Q2. when applied to convolutional layers the hierarchy effectively imposes a spatial structure to grouping neighboring pixels. Could you show those patterns? would it make sense to shuffle the column vectors resulting from im2col to control these patterns?
> - A2. The densification flow somehow shows this already. If we shuffle the im2col it means we are NOT sampling the receptive field in sequence, so it may not be do-able.
>
> - Q3. In the context of compression, many other methods exist but they're not shown in the paper.
> - A3. Yes, that is because we choose the approaches (FastFood and standard Butterfy) in the same category as DeBut to make the comparison. For low-rank factorization methods (such as tensor decomposition or SVD approaches), they all exploit low-rank matrix or tensor decompositions, but the factors in DeBut/FastFood/Butterfly all have full ranks, and consequently do not belong to the class of low-rank decomposition techniques. However, we emphasize that compared with the low-rank decomposition approaches, DeBut can have higher accuracy with fewer parameters as shown in the table below. For pruning and weight quantization approaches, we remark that DeBut is orthogonal and complementary to them and can be easily plugged in to compress the model further. Therefore, we do not compare DeBut with pruning and weight quantization approaches.
>
> |   Model   |   Method   |  Accuracy(%)  |    #params    |
> | ----------------| ----------------- | --------------------- | -------------------- |
> |  VGG16   |   Baseline  |    93.96       |  14.99M(0.0%) |
> |  VGG16   |   Tucker2 [1]  |    93.36      |  3.21M(78.58%) |
> |  VGG16   |   DeBut    |    93.71      |  2.43M(83.77%) |
>
> [1] Kim, Yong-Deok, et al. "Compression of deep convolutional neural networks for fast and low power mobile applications."

---

### Official Review · Reviewer_Nvoh · 2021-07-16

**Rating:** 6
**Confidence:** 3

**Summary:**

The paper proposes "Deformable Butterfly" (DeBUT) a generalization of butterfly transforms (the linear operator behind the FFT), which can support non power-of-to inputs/outputs.
It studies how to plug such transforms into various architectures (LeNet, VGG, ResNet), from the perspective of filter design, initialization, parameter savings and accuracy loss.
Overall it is demonstrated that DeBut can achieve a favourable tradeoff compared to previous butterfly-style approaches such as "Buterfly" [5,6] and "Adaptive Fastfood" [35].


**Main Review:**

Overall the paper is very well written, with clear figures demonstrating the concepts of interests. The
presented generalization of butterfly matrices seems very natural, and addresses the following challenges:

- How to ensure that there are no "dead parts" in the chain of linear transforms.
- How to go increase or decrease the number of channels (or doing both in so called "bulging")
- How to initialize the transforms with an alternative least squares (ALS) method
- How to fine tune the resulting network

The experiments are relatively extensive, and the results in Tab 4.-6. are favorable to the proposed method.

My main concerns are the following:

- The relationship between the generalized butterfly transform and generalized FFT.
 It is well known the FFT (e.g. the Cooley–Tukey FFT algorithm) can recurse over any factorization of the input size, not just powers of two (POT). Is there a relationship between this and the proposed DeBUT transforms?
- The design of the transforms seems rather complicated. It would be more convincing if the authors could present a simple rule on how to transform layers.
- Related: there is a constraint p/r = q/s that needs to be satisfied. This implies the channels need to be prime factorizable if you want to transform them. I don't see any discussion about this.

- Inference times are presented in Tab. 6, without any details. Is this exploiting the structure of the transform? Why is the baseline (original network) missing from the benchmark?

- Does not seem to work without ALS initialization for large networks? I did not find the results for this, although they were said to be found in Table A3. in the appendix.

- in l.104 it is said DeBut "replaces" the circulant structure of CNN convolution. This sounds wrong to me, as the doubly circulant structure emerges from the same matrix multiplication being applied to to each patch extracted from the im2col operation. DeBut is also doing this, so the circulant structure remains (you could say with a nested butterfly structure).


Final update: Given the rebuttal and the other positive reviews, I am keeping my score.


**Time Spent Reviewing:**

2

---

> ### Author Response · Authors · 2021-08-10
> **Thanks for the very concise summary of DeBut**
>
> Thanks for the very concise summary of DeBut, especially on its natural extension (relaxation) from Butterfly matrices to adapt to different I/O dimensions, so that there is no wastage in the "dead parts" as mentioned.
>
> - Q1. The relationship between the generalized butterfly transform and generalized FFT. It is well known the FFT (e.g. the Cooley–Tukey FFT algorithm) can recurse over any factorization of the input size, not just powers of two (POT). Is there a relationship between this and the proposed DeBUT transforms?
> - A1. Yes. Some generalized FFT allows for divide-and-conquer of any size that divides n, and it is most useful when n is divisible by a series of prime factors. From this point of view, the proposed DeBut is the same since it needs small prime factors to generate the chains. Because the standard butterfly can be regarded as a special case of DeBut, DeBut is expected to have a more substantial ability to learn an efficient algorithm for many important transforms, including the generalized FFT.
>
> - Q2. The design of the transforms seems rather complicated. It would be more convincing if the authors could present a simple rule on how to transform layers.
> - A2.  In brief, we always design a DeBut chain from the right (factor with t=1) to the left (factor only has one block) considering the following three aspects: 1) monotonic or bulging chain, 2) low or high compression rate, 3) fast or slowly shrinking information flow.
> When we design the rightmost factor, t is always equal to 1 to pass the bipolar (information flow) test. Once we decide p of the factor on the left side, we set r and s, taking into account the three points mentioned above. Step by step, we get the completed chain at the end.
> Indeed, we have written a chain generation demo to showcase how to generate chains automatically in detail. Please check the anonymous Github: https://anonymous.4open.science/r/Rebuttal-773-6DF6
>
> - Q3. Related: there is a constraint p/r = q/s that needs to be satisfied. This implies the channels need to be prime factorizable if you want to transform them. I don't see any discussion about this.
> - A3. Sorry for causing confusion about how to design the chains for layers with different \#channels. We did not discuss this in detail since most neural networks set the \#channels to be factorizable like 16, 64, 128, and 512, etc., where the constraint p/r = q/s can be easily satisfied. We use the padding trick to solve the problem for the particular case that the #channels is not factorizable. For example, if the flattened weight matrix is of size [217, 509], we can use zeros to expand the matrix to size [256, 512] and then apply the DeBut technique on the newly obtained matrix. It is worth noting that the additional elements in the first and the last factors will be discarded after the ALS initialization. Please kindly refer to the demo provided in Q2, in which we have shown an example about the special case. We will definitely add discussion about different #channels cases to the accepted paper.
>
> - Q4. Inference times are presented in Tab. 6, without any details. Is this exploiting the structure of the transform? Why is the baseline (original network) missing from the benchmark?
> - A4. In Table 6, we compare the performance of Adaptive FastFood, Butterfly, and DeBut by substituting the selected CONV layers with different linear structures. The results reported of Adaptive FastFood and Butterfly are our implementation using their officially released codes. For DeBut, the chains for the corresponding layers are the same as described in Table 3. We will add the explanation to the caption of Table 6 to make it easier to understand. The original VGG-16-BN's accuracy and training/testing time are 93.96%, 48s/epoch, and 2.01s, respectively. We want to stress that the training/testing of the original VGG-16 is faster than DeBut, Butterfly, and FastFood because the present PyTorch implementation of the linear transform approaches contains lots of loops, which can be further optimized. In other words, we want to note that the comparison of DeBut, Butterfly, and FastFood is still fair since all of them are implemented in standard coding and are not optimized at the algorithmic/hardware level.
>
> - Q5. Does not seem to work without ALS initialization for large networks? I did not find the results for this, although they were said to be found in Table A3. in the appendix.
> - A5. In Table A3, the accuracy with and without brackets are results of w/o and w/ ALS scheme, respectively. We omit the w/o ALS case in Table 3 to focus on the performance of w/ ALS case. Based on our experimental results, w/ ALS has better performance for larger layers, so we directly choose the better w/ ALS scheme for ResNet-50, and report the results in Table 4. We will add back the w/o ALS results from the Appendix to the accepted paper to make the comparison between w/ and w/o ALS more intuitive to the readers.
>
> - Q6. in l.104 it is said DeBut "replaces" the circulant structure of CNN convolution. This sounds wrong to me, as the doubly circulant structure emerges from the same matrix multiplication being applied to to each patch extracted from the im2col operation. DeBut is also doing this, so the circulant structure remains (you could say with a nested butterfly structure).
> - A6. Not really, the doubly circulant does not conform to the im2col. In im2col, the flattened filter matrix is F does not have the circulant structure. The circulant CNN matrix appears when we simply vectorize the input feature instead of the patch-wise fashion in im2col.

---

### Official Review · Reviewer_fZAT · 2021-07-17

**Rating:** 6
**Confidence:** 4

**Summary:**

In this work, the authors propose a new way named deformable butterfly (DeBut) to represent the linear transformation between feature layers in a deep neural network. It is based on a chain of butterfly factor matrices that can be used together to form a densified matrix for linear transformation. The experiments show the proposed method can reduce the model size while maintaining similar test accuracy.

**Limitations And Societal Impact:**

The limitations should be better explained, and I do not see negative societal impact.

**Main Review:**

Strengths:
- Compared to the prior work [5,6], there is an extension from square matrix to non-square matrix for linear transformation.
- The runtime complexity can be lower than O(nlogn).
- The number of parameters can be reduced.
- The idea of DeBut looks quite novel to me.
- The paper is well-written and the complicated idea is well illustrated with figures and explanations.

Weaknesses:
- It seems that the accuracy may drop a little bit with DeBut, although the model size is reduced. In Table 4, the Top-1 accuracy is lower than ResNet-50.
- Is there any example to demonstrate the difference DeBut-mono and DeBut-bulging? I am confused by these two designs. Why the authors propose these two designs and what are the pros and cons?
- Are the results shown in the experiments by training a model with DeBut from scratch (or fine-tuning) or approximating a pre-trained model with DeBut?
- Why DeBut does not improve the model accuracy, compared to Conv and FC?
- It seems the model with DeBut runs slower than the baseline (e.g., ResNet50)?





**Time Spent Reviewing:**

2

---

> ### Author Response · Authors · 2021-08-10
> **We sincerely appreciate your positive comments on DeBut**
>
> We sincerely appreciate your positive comments on DeBut which we very much like to introduce to the community. Indeed, we have obtained further interesting results between paper submission and now, such as the BNN variant of DeBut.
>
> - Q1. It seems that the accuracy may drop a little bit with DeBut, although the model size is reduced. In Table 4, the Top-1 accuracy is lower than ResNet-50.
> - A1. Yes, by the "no free lunch" theorem, reducing the number of parameters would normally lead to a drop of Top-1 accuracy as is common in most other compression methods (e.g., pruning, sparsification, low-rank decomposition etc.).
>
> - Q2. Is there any example to demonstrate the difference DeBut-mono and DeBut-bulging? I am confused by these two designs. Why the authors propose these two designs and what are the pros and cons?
> - A2. The reason of having both types of chains is to demonstrate the "deformable" geometry one can have for the chains. Through our experiments, w/ and w/o ALS, it is found that both types of chains have their merits in specific problems. We can compare the two types of chains from the following aspects: 1) Compression ability: a monotonic chain can obtain a more compact model than a bulging chain since the expanding factor in the bulging chain introduces additional parameters. 2) Accuracy: a bulging chain is expected to have a higher accuracy since in principle it has a higher representation power, and may learn better latent information, though not necessarily always the case. 3) Stability: we observed that the variance of the accuracy obtained by a monotonic chain is smaller than that obtained by a bulging chain. In summary, the bulging chain is the first choice if higher accuracy is desired, while monotonic chains are preferred when a more robust model with much fewer parameters is required. Indeed, we have written a chain generation demo to demonstrate how to generate chains automatically in detail. Please check the anonymous Github: https://anonymous.4open.science/r/Rebuttal-773-6DF6
>
> - Q3. Are the results shown in the experiments by training a model with DeBut from scratch (or fine-tuning) or approximating a pre-trained model with DeBut?
> - A3. We have tested both, w/o ALS means training DeBut chains from scratch (viz. random initialization), while w/ ALS means we initialize the factors in a DeBut chain using ALS so that the chain’s product is close to that particular layer in a pretrained model.
>
> - Q4. Why DeBut does not improve the model accuracy, compared to Conv and FC?
> - A4. DeBut is the sparsification of Conv and FC linear transform by structured matrix decomposition, with much reduced number of parameters. Consequently, we do expect a slight drop in accuracy as is common across almost every neural network compression method. That said, we want to report one interesting experiment: We substitute the last CONV layer in VGG16 using two very similar chains with different dense blocks in the rightmost factor, namely, 2x2 and 8x9, respectively. Except for the substituted last layer, other layers are all initialized with a pretrained VGG16 on CIFAR-10. It is worth noting that the accuracy of the chain with 8x9 dense blocks in the rightmost factor reaches 94.11%, which is higher than the chain with 2x2 dense blocks in the rightmost factor (as in a standard butterfly) and even higher than the baseline. This observation echoes the property that the dense blocks in the rightmost factor in a DeBut chain can conform to the kernel size to give rise to a higher system accuracy.
>
> |     Model      | Rightmost block | #params | Accuracy(%) |
> | ---------------------- | -------------------- | ----------- | ---------------- |
> | VGG16(Baseline) |     N/A      | 14.99M  |   93.96    |
> | VGG16         |     2*2      | 12.71M  |   93.92    |
> | VGG16         |     8*9      | 12.71M  |   94.11    |
>
>
> - Q5. It seems the model with DeBut runs slower than the baseline (e.g., ResNet50)?
> - A5. We use an analogy. For example, in principle the WHT (Walsh Hadamard Transform) should have a lower complexity than FFT, but our experience is that cuFFT is around five times faster than the direct implementation of WHT due to the highly specialized and optimized implementation of the former. That said, in our tests, all algorithms are compared in the standard coding and software environment, so that the comparison is still fair.
> In short, we want to note that DeBut is done in standard coding and is not optimized in the algorithmic/hardware level (which we will pursue as a future work). But borrowing from the experience of cuFFT, we believe specialized implementation of DeBut will be similarly efficient!

---

### Author Response · Authors · 2021-08-18
**DeBut can further prune the already pruned!**

We were inspired to check whether DeBut can be applied to a pruned CNN model to further compress its number of network weights. For a quick test, we ran it on a channel-pruned VGG16 trained on CIFAR10 [1], and the result turned out to be amazing! Namely, the DeBut replacement achieves a higher accuracy even with more parameters reduced, captured in the table below.

| Model              | # Params | Acc    |
| ------------------ | -------- | ------ |
| VGG16(baseline)    | 14.99M   | 93.96% |
| VGG16(HRank[1])       | 2.76M    | 93.73% |
| VGG16(HRank+DeBut) | 1.67M    | 93.82% |

[1] Lin, Mingbao, et al. "Hrank: Filter pruning using high-rank feature map." CVPR 2020.

Without delving into details, the I/O channel numbers of the pruned CNN are rather rugged (of course, far from being powers-of-two) and yet unlike existing PoT methods, DeBut can be easily adapted to fit them all (example codes will be released upon publication of the paper). Subsequently, this test further validates and showcases the capability of DeBut in achieving additional compression without harming accuracy.

Indeed, we believe for NeurIPS the novelty of an idea (not just its performance) and the insights & inspiration it can bring for further exploration are crucial, which is the case for DeBut. For one thing, non-square FC layers are heavily present in LSTM & Transformers, where DeBut can definitely help.

Again, we sincerely thank all the reviewers for all their useful comments which we will definitely work into the final version. Your favorable consideration of our work is highly appreciated.

---

### Author Response · Authors · 2021-08-31
**Seeing how a Debut sees**

In recent days we have unveiled more inspiring insights and connection of a Deformable Butterfly (DeBut) chain to a CNN-like interpretation. We think they may be of interest to the AC and reviewers and can be useful to further justify the theoretical and practical values of DeBut.

We use a toy example for illustration. Suppose we have a 3x3x2 (i.e. 2-channel) input tensor whose front and rear slices are
$ \\begin{bmatrix}
1 & 4 & 7 \\\\
2 & 5 & 8 \\\\
3 & 6 & 9
\\end{bmatrix}  $ and $ \\begin{bmatrix}
10 & 13 & 16 \\\\
11 & 14 & 17 \\\\
12 & 15 & 18
\\end{bmatrix}  $, respectively.

Without zero padding, we apply eight 2x2x2 filters (stride=1) in a column-major manner to produce a 2x2x8 (i.e. 8-channel) output tensor. With reference to Fig. 2 in our paper, using im2col to transform the input tensor into an 8x4 input matrix (viz. we have 4 pixels in each output channel slice), and turning the 8x8 flattened filter matrix F into a Butterfly chain results in

$
\\begin{bmatrix}
\* &  &  &  & \* &  &  &  \\\\
 & \* &  &  &  & \* &  &  \\\\
 &  & \* &  &  &  & \* &  \\\\
 &  &  & \* &  &  &  & \* \\\\
\* &  &  &  & \* &  &  &  \\\\
 & \* &  &  &  & \* &  &  \\\\
 &  & \* &  &  &  & \* &  \\\\
 &  &  & \* &  &  &  & \*
\\end{bmatrix}
$ $ \\begin{bmatrix}
\* &  & \* &  &  &  &  &  \\\\
 & \* &  & \* &  &  &  &  \\\\
\* &  & \* &  &  &  &  &  \\\\
 & \* &  & \* &  &  &  &  \\\\
 &  &  &  & \* &  & \* &  \\\\
 &  &  &  &  & \* &  & \* \\\\
 &  &  &  & \* &  & \* &  \\\\
 &  &  &  &  & \* &  & \*
\\end{bmatrix}  $ $ \\begin{bmatrix}
\* & \* &  &  &  &  &  &  \\\\
\* & \* &  &  &  &  &  &  \\\\
 &  & \* & \* &  &  &  &  \\\\
 &  & \* & \* &  &  &  &  \\\\
 &  &  &  & \* & \* &  &  \\\\
 &  &  &  & \* & \* &  &  \\\\
 &  &  &  &  &  & \* & \* \\\\
 &  &  &  &  &  & \* & \*
\\end{bmatrix}  $ $ \\begin{bmatrix}
1 & 2 & 4 & 5 \\\\
2 & 3 & 5 & 6 \\\\
4 & 5 & 7 & 8 \\\\
5 & 6 & 8 & 9 \\\\
\hline
10 & 11 & 13 & 14 \\\\
11 & 12 & 14 & 15 \\\\
13 & 14 & 16 & 17 \\\\
14 & 15 & 17 & 18
\\end{bmatrix}  $

Now, if we pick the first output channel (i.e., the top row of the matrix chain product) and see how the information aggregates through the chain by marking the matrix entries involved with a “+” sign below:

$
\\begin{bmatrix}
\+ &  &  &  & \+ &  &  &  \\\\
 & \* &  &  &  & \* &  &  \\\\
 &  & \* &  &  &  & \* &  \\\\
 &  &  & \* &  &  &  & \* \\\\
\* &  &  &  & \* &  &  &  \\\\
 & \* &  &  &  & \* &  &  \\\\
 &  & \* &  &  &  & \* &  \\\\
 &  &  & \* &  &  &  & \*
\\end{bmatrix}
$ $ \\begin{bmatrix}
\+ &  & \+ &  &  &  &  &  \\\\
 & \* &  & \* &  &  &  &  \\\\
\* &  & \* &  &  &  &  &  \\\\
 & \* &  & \* &  &  &  &  \\\\
 &  &  &  & \+ &  & \+ &  \\\\
 &  &  &  &  & \* &  & \* \\\\
 &  &  &  & \* &  & \* &  \\\\
 &  &  &  &  & \* &  & \*
\\end{bmatrix}  $ $ \\begin{bmatrix}
\+ & \+ &  &  &  &  &  &  \\\\
\* & \* &  &  &  &  &  &  \\\\
 &  & \+ & \+ &  &  &  &  \\\\
 &  & \* & \* &  &  &  &  \\\\
 &  &  &  & \+ & \+ &  &  \\\\
 &  &  &  & \* & \* &  &  \\\\
 &  &  &  &  &  & \+ & \+ \\\\
 &  &  &  &  &  & \* & \*
\\end{bmatrix}  $ $ \\begin{bmatrix}
1 & 2 & 4 & 5 \\\\
2 & 3 & 5 & 6 \\\\
4 & 5 & 7 & 8 \\\\
5 & 6 & 8 & 9 \\\\
\hline
10 & 11 & 13 & 14 \\\\
11 & 12 & 14 & 15 \\\\
13 & 14 & 16 & 17 \\\\
14 & 15 & 17 & 18
\\end{bmatrix}  $

then we can observe on the right that the operation is indeed 2x1 vector kernels acting on the input slices in a depthwise convolution manner, followed by mixing among the convolved input channels to eventually produce the first output channel. To give one more example, now we repeat with the second output channel:

$
\\begin{bmatrix}
\* &  &  &  & \* &  &  &  \\\\
 & \+ &  &  &  & \+ &  &  \\\\
 &  & \* &  &  &  & \* &  \\\\
 &  &  & \* &  &  &  & \* \\\\
\* &  &  &  & \* &  &  &  \\\\
 & \* &  &  &  & \* &  &  \\\\
 &  & \* &  &  &  & \* &  \\\\
 &  &  & \* &  &  &  & \*
\\end{bmatrix}
$ $ \\begin{bmatrix}
\* &  & \* &  &  &  &  &  \\\\
 & \+ &  & \+ &  &  &  &  \\\\
\* &  & \* &  &  &  &  &  \\\\
 & \* &  & \* &  &  &  &  \\\\
 &  &  &  & \* &  & \* &  \\\\
 &  &  &  &  & \+ &  & \+ \\\\
 &  &  &  & \* &  & \* &  \\\\
 &  &  &  &  & \* &  & \*
\\end{bmatrix}  $ $ \\begin{bmatrix}
\* & \* &  &  &  &  &  &  \\\\
\+ & \+ &  &  &  &  &  &  \\\\
 &  & \* & \* &  &  &  &  \\\\
 &  & \+ & \+ &  &  &  &  \\\\
 &  &  &  & \* & \* &  &  \\\\
 &  &  &  & \+ & \+ &  &  \\\\
 &  &  &  &  &  & \* & \* \\\\
 &  &  &  &  &  & \+ & \+
\\end{bmatrix}  $ $ \\begin{bmatrix}
1 & 2 & 4 & 5 \\\\
2 & 3 & 5 & 6 \\\\
4 & 5 & 7 & 8 \\\\
5 & 6 & 8 & 9 \\\\
\hline
10 & 11 & 13 & 14 \\\\
11 & 12 & 14 & 15 \\\\
13 & 14 & 16 & 17 \\\\
14 & 15 & 17 & 18
\\end{bmatrix}  $

Again, it is a vector kernel filtering action on the right followed by mixing of information along the chain towards the left.
As such, there is a theoretical insight of DeBut being an application of vector kernels to the input tensor, whose effectiveness agrees with the finding in [1]. Nonetheless, Ref. [1] is still just standard CNN, while DeBut differs in that there is additional mixing of information down the chain instead of just one-step CNN filtering.

Moreover, in a CNN, 3x3 kernels almost always work better than 2x2 kernels, and therefore we need the [arbitrary integer x3] dense blocks in the rightmost DeBut factor to play the role of doing 3x1 vector kernels, which cannot be achieved via standard Butterfly or Fastfood Transform. We hope this sheds light on explaining the excellent performance of DeBut in replacing the CNN layers.

[1] J. Ou & Y. Li, “Vector-kernel convolutional neural networks”, Neurocomputing, 2019.

---

### Decision · Program_Chairs · 2021-09-28

**Decision:**

Accept (Poster)

**Comment:**

Congratulations, the paper is accepted to NeurIPS 2021!
Please incorporate the edits and corrections as discussed in the rebuttal and reviews.
Furthermore, tone down the "sensational" language used in the work; try to keep language as factual as possible.

**Consistency Experiment:**

NeurIPS has a long history of experimentation. In 2014, NeurIPS ran an experiment in which 10% of submissions were reviewed by two independent committees to quantify the randomness in the review process. This year, we repeated a variant of this experiment to see how the quality of the review process has changed over time.  This paper was part of the experiment and was therefore assigned to two committees (consisting of reviewers, an Area Chair, and a Senior Area Chair) that reached independent decisions.  If both committees made the same recommendation, this recommendation was followed. If a single committee recommended acceptance, the paper was accepted (with the exception of a few cases in which the other committee identified what we considered a fatal flaw, e.g., an error in a key result).

Both committees reached the same decision: **Accept (Poster)**

The other committee assigned to the paper recommended **Accept (Poster)**.  You can find the other set of reviews, along with any follow up discussion with the authors here:
https://openreview.net/forum?id=ogjTzvtqbtK